# Interacting tipping elements increase risk of climate domino effects under global warming

Nico Wunderling[1,2,3], Jonathan F. Donges[1,4], Jürgen Kurths[1,5], and Ricarda Winkelmann[1,2]

[1]Earth System Analysis and Complexity Science, Potsdam-Institute for Climate Impact Research (PIK), Member of the Leibniz Association, 14473 Potsdam, Germany
[2]Institute of Physics and Astronomy, University of Potsdam, 14476 Potsdam, Germany
[3]Department of Physics, Humboldt University of Berlin, 12489 Berlin, Germany
[4]Stockholm Resilience Centre, Stockholm University, Stockholm, SE-10691, Sweden
[5]Saratov State University, Saratov, RU-410012, Russia

**Correspondence:** Nico Wunderling (nico.wunderling@pik-potsdam.de), Ricarda Winkelmann (ricarda.winkelmann@pik-potsdam.de)

**Abstract.** With progressing global warming, there is an increased risk that one or several tipping elements in the climate system might cross a critical threshold, resulting in severe consequences for the global climate, ecosystems and human societies. While the underlying processes are fairly well understood, it is unclear how their interactions might impact the overall stability of the Earth's climate system. As of yet, this cannot be fully analysed with state-of-the-art Earth system models due to computational constraints as well as some missing and uncertain process representations of certain tipping elements. Here, we explicitly study the effects of known physical interactions among the Greenland and West Antarctic ice sheets, the Atlantic Meridional Overturning Circulation (AMOC) and the Amazon rainforest using a conceptual network approach. We analyse the risk of domino effects being triggered by each of the individual tipping elements under global warming in equilibrium experiments. In these experiments, we propagate the uncertainties in critical temperature thresholds, interaction strengths and interaction structure via large ensembles of simulations in a Monte-Carlo approach. Overall, we find that the interactions tend to destabilise the network of tipping elements. Furthermore, our analysis reveals the qualitative role of each of the four tipping elements within the network, showing that the polar ice sheets on Greenland and West Antarctica are oftentimes the initiators of tipping cascades, while the AMOC acts as a mediator transmitting cascades. This indicates that the ice sheets, which are already at risk of transgressing their temperature thresholds within the Paris range of 1.5 to 2 °C, are of particular importance for the stability of the climate system as a whole.

## 1   Introduction

### 1.1   Tipping elements in the climate system

The Earth system comprises a number of large-scale subsystems, the so-called *tipping elements*, that can undergo large and possibly irreversible changes in response to environmental or anthropogenic perturbations once a certain critical threshold in

forcing is exceeded (Lenton et al., 2008). Once triggered, the actual tipping process might take several years up to millennia depending on the respective response times of the system (Hughes et al., 2013; Lenton et al., 2008). Among the tipping elements are cryosphere components such as the continental ice sheets on Greenland and Antarctica, biosphere components such as the Amazon rainforest, boreal forests and coral reefs as well as large-scale atmospheric and oceanic circulation patterns such as monsoon systems and the Atlantic Meridional Overturning Circulation (AMOC). With continuing global warming, it becomes more likely that critical thresholds of some tipping elements might be exceeded, possibly within this century, triggering severe consequences for ecosystems, infrastructures and human societies. These critical thresholds can be quantified with respect to the global mean temperature (GMT), resulting in three clusters of tipping elements that are characterised by their critical temperature between 1–3 °C, 3–5 °C, and above 5 °C of warming compared to pre-industrial temperatures, respectively (Schellnhuber et al., 2016). The most vulnerable cluster, which is already at risk between 1–3 °C of warming, includes several cryosphere components, specifically mountain glaciers as well as the Greenland and West Antarctic ice sheets. Recent studies suggest potential early-warning indicators for these tipping elements, showing that some of them are approaching or might have already transgressed a critical threshold (Lenton et al., 2019; Caesar et al., 2018; Nobre et al., 2016; Favier et al., 2014).

## 1.2 Interactions between climate tipping elements

The tipping elements in the Earth system are not isolated systems, but interact on a global scale (Lenton et al., 2019; Kriegler et al., 2009). These interactions could have stabilising or destabilising effects, increasing or decreasing the probability of emerging tipping cascades, and it remains an important problem to understand how this affects the overall stability of the Earth system. Despite the considerable recent progress in global Earth system modelling, current state-of-the-art Earth system models cannot yet comprehensively simulate the nonlinear behaviour and feedbacks between some of the tipping elements due to computational limitations (Wood et al., 2019). Furthermore, the interactions between tipping elements have only partially been described in a framework of more conceptual, but process-based models, and our current understanding of the interaction structure of tipping elements is partly based on expert knowledge. For a subset of five tipping elements, an expert elicitation was conducted synthesising a causal interaction structure and an estimation for the probability of tipping cascades to emerge (Kriegler et al., 2009). These studied tipping elements were the Greenland Ice Sheet, the West Antarctic Ice Sheet, the Atlantic Meridional Overturning Circulation (AMOC), the El-Niño Southern Oscillation (ENSO) and the Amazon rainforest (see Fig. 1 and Fig. S3). Although this network is not complete with respect to the physical interactions between the tipping elements and the actual set of tipping elements themselves (Wang & Hausfather, 2020; Lenton et al., 2019; Steffen et al., 2018), it presented a first step towards synthesising the positive and negative feedbacks between climate tipping elements. To our best knowledge, a systematic update of this assessment or a comparably comprehensive expert assessment has not been undertaken since Kriegler et al. (2009).

Based on the network from this expert elicitation and a Boolean approach based on graph grammars, an earlier study found that the strong positive-negative feedback loop between the Greenland Ice Sheet and the AMOC might act as a stabiliser to the

Earth system (Gaucherel & Moron, 2017). Also, using the interaction network data of Kriegler et al. (2009), it has been shown that large economic damages due to tipping cascades could arise with respect to the social cost of carbon, using a stochastic and dynamic evaluation of tipping points in an integrated assessment model (Cai et al., 2016). Other studies also quantified the economic impacts of single climate tipping events and tipping interactions (Lemoine & Traeger, 2016; Cai et al., 2015). In the light of recent studies that hypothesise a considerable risk of current anthropogenic pressures triggering tipping cascades, up to a potential global cascade (towards a so-called "hothouse state" of the Earth system) (Lenton et al., 2019; Steffen et al., 2018), we here aim at developing a conceptual dynamic network model that can assess whether interactions of tipping elements have an overall stabilising or destabilising effect on the global climate state. As such, we view our approach as an hypotheses generator that produces qualitative scenarios (rather than exact quantifications or projections) that can then be further examined by more process-detailed Earth system models. In this way, the results of this study can lay the foundations and possibly guide towards a more detailed analysis with more complex models or data-based approaches.

## 1.3   Constraints from current observations and paleoclimatic evidence

Observations over the past decades show that several tipping elements are already impacted by progressing global warming (Wang & Hausfather, 2020; Lenton et al., 2019; IPCC, 2014; Levermann et al., 2010). Ice loss from Greenland and West Antarctica has increased and accelerated over the past decades (Shepherd et al., 2018; Khan et al., 2014; Zwally et al., 2011). Recent studies suggest that the Amundsen basin in West Antarctica might in fact have already crossed a tipping point (Favier et al., 2014; Rignot et al., 2014). The grounding lines of glaciers in this region are rapidly retreating, which could induce the Marine Ice Sheet Instability and eventually lead to the disintegration of the entire basin (Mercer, 1978; Weertman, 1974). Paleoclimate records suggest that parts of Antarctica and larger parts of Greenland might already have experienced strong ice retreat in the past, especially during the Pliocene as well as during Marine Isotope Stages 5e and 11 (Dutton et al., 2015).

It has also been shown that the AMOC experienced a significant slow-down since the mid 20th century (Caesar et al., 2018), potentially due to freshening of the North Atlantic ocean by increased meltwater influx from Greenland (Bakker et al., 2016; Böning et al., 2016). An AMOC slow-down has likely also occurred during the last deglaciation in the Heinrich event 1 and Younger Dryas cold periods, as proxies from sea surface and air temperatures as well as climate model simulations suggest (Ritz et al., 2013).

The Amazon rainforest is not only directly impacted by anthropogenic climate change, including the increased risk of extensive drought events or heat waves (Marengo et al., 2015; Brando et al., 2014), but also by deforestation and fire (Thonicke et al., 2020; Malhi et al., 2009). This increases the likelihood that parts of it will shift from a rainforest to a savannah state for instance through diminished moisture recycling (Staal et al., 2018; Zemp et al., 2017). It is suspected that the Amazon rainforest could be close to a critical extent of deforestation which might, together with global warming, suffice to initiate such a critical transition (Nobre et al., 2016). This could put 30–50% of rainforest ecosystems at risk of shifting the rainforest to

tropical savannah or dry forests (Nobre et al., 2016). On a local to regional point of view, the potential for critical transitions in

the rainforest is further examined by more recent studies (Staal et al., 2020; Ciemer et al., 2019).

## 1.4   Structure of this work

Following the introduction, in Sect. 2, we provide an overview of the biogeophysical processes governing the individual

dynamics and interactions of the four tipping elements considered here, and how these are represented in our conceptual

network model. We also describe the construction of the large-scale Monte-Carlo ensemble which enables us to propagate the

parameter uncertainties inherent in the modelled tipping elements and their interactions. In Sect. 3, we explore how the critical

threshold temperature ranges of the tipping elements change with increasing overall interaction strength. It is also shown which

tipping elements initiate and transmit tipping cascades, revealing the characteristic roles of the tipping elements in the Earth

system. We also discuss the distinct nature of ENSO as a potential tipping element, and present results of a robustness analysis

including this additional tipping element in our network model. Sect. 4 summarises the results and discusses the limitations of

our approach. It also outlines possible further lines of research concerning tipping element interactions and risks of emerging

tipping cascades with more process-detailed models.

## 2   Methods

In the following, we present our dynamic network approach for modelling tipping interactions and cascades in the Earth

system. In Sect. 2.1, we motivate the use of a stylised equation to represent climate tipping elements in a conceptual manner.

This equation exhibits a double-fold bifurcation (see Fig. 2)

$$\frac{dx_i}{dt} = \left[ -x_i^3 + x_i + c_i \right] \frac{1}{\tau_i}. \tag{1}$$

Here, $x_i$ indicates the state of a certain tipping element, $c_i$ is the critical parameter and $\tau_i$ the typical tipping time scale with $i =$

{Greenland Ice Sheet, West Antarctic Ice Sheet, AMOC, Amazon rainforest}. This approach has already been used frequently

for qualitatively describing tipping dynamics in different applications and network types and has been applied to systems in

climate, ecology, economics and political science (Klose et al., 2020; Krönke et al., 2020; Wunderling et al., 2020a; Dekker et

al., 2018; Brummitt et al., 2015; Abraham et al., 1991).

To describe the tipping elements' interactions, we extend Eq. 1 by a linear coupling term (Klose et al., 2020; Krönke et al.,

2020; Brummitt et al., 2015) to yield

$$115 \quad \frac{dx_i}{dt} = \left[ \overbrace{-x_i^3 + x_i + c_i}^{\text{Individual dynamics term}} + \overbrace{\frac{1}{2} \sum_{\substack{j \\ j \neq i}} d_{ij} \left( x_j + 1 \right)}^{\text{Coupling term}} \right] \frac{1}{\tau_i}, \tag{2}$$

and describe the physical interpretation of these interactions between the tipping elements in Sect. 2.2. While the first term

(*individual dynamics term*) determines the dynamical properties of each individual tipping element, the second term (*coupling*

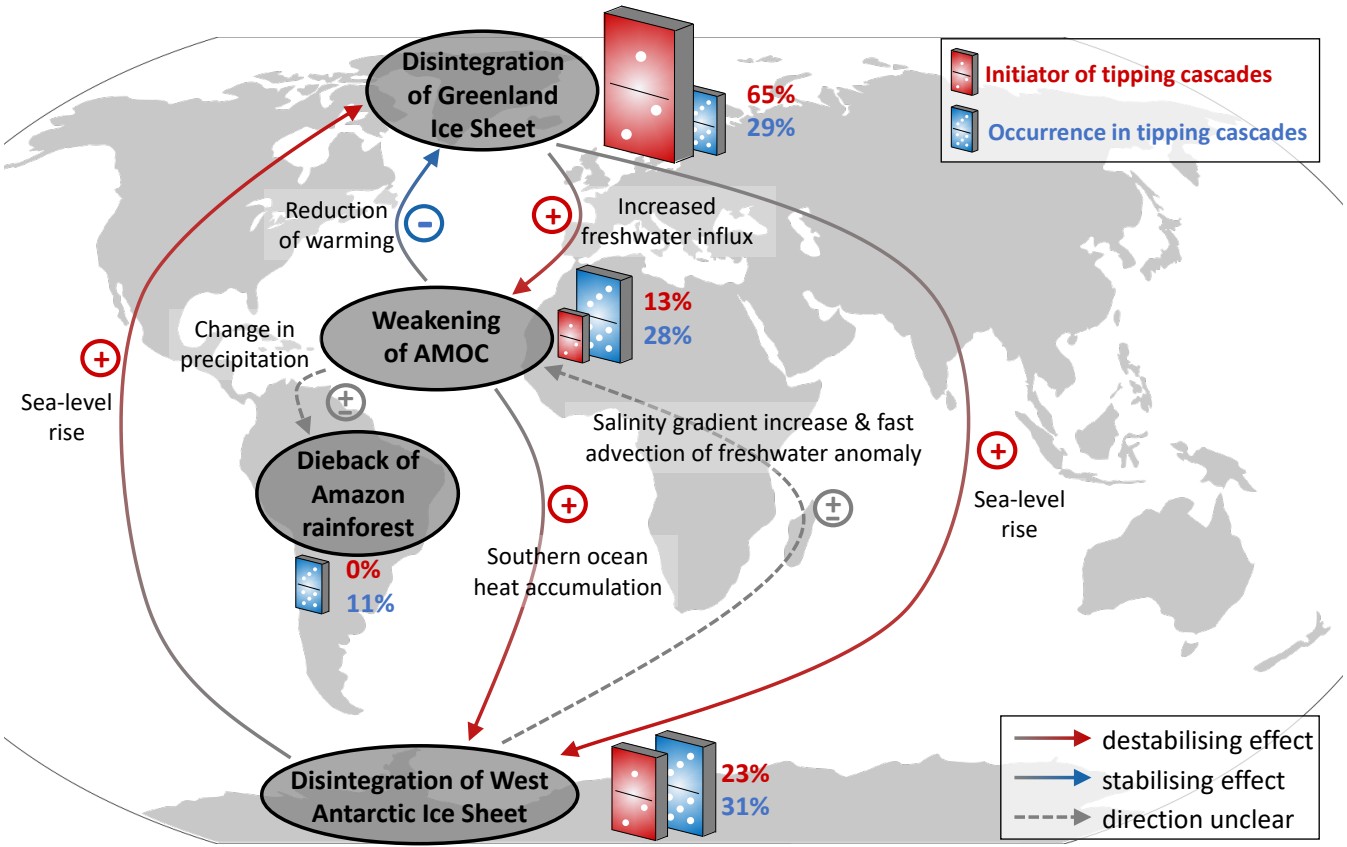

**Figure 1.** Interactions between climate tipping elements and their roles in tipping cascades. The Greenland Ice Sheet, West Antarctic Ice Sheet, Atlantic Meridional Overturning Circulation (AMOC) and the Amazon rainforest are depicted together with their main interactions (Kriegler et al., 2009). The links between the tipping elements are colour-coded, where red arrows depict destabilising and blue arrows depict stabilising interactions. Where the direction is unclear, the link is marked in grey. A more thorough description of each of the tipping elements and the links can be found in Tables 1, 2 and in Sect. 2. Where tipping cascades arise, the relative size of the dominoes illustrates in how many ensemble members the respective climate component initiates (red domino) or occurs in tipping cascades (blue domino). Standard deviations for these values are given in Figs. S1(a) and (b). Generally, the polar ice sheets are found to more frequently take on the role as initiators of cascades than the AMOC and Amazon rainforest.

*term*) describes the effects of interactions between tipping elements. If the prefactors in front of the cubic and the linear term are unity as in Eq. 2 and the additive coupling term is neglected ($d_{ij} = 0$ for all $i$, $j$), the critical threshold values where qualitative state changes occur are $c_{i\,1,2} = \pm\sqrt{4/27}$ (Klose et al., 2020). The system described by this differential equation is bistable for values of the critical parameter between $c_1$ and $c_2$ and can here be separated into a *transitioned* and a *baseline* state, where $x_i = -1$ denotes the baseline state and $x_i = +1$ the completely transitioned one (see Fig. 2).

Building on these model equations, in Sect. 2.3, we describe the fully parameterised model and its parameters as it is used in this study. Specifications of how tipping cascades are evaluated and time scales are chosen can be found in Sects. 2.4 and 2.5. Lastly, our large scale Monte Carlo ensemble approach for the propagation of parameter and interaction network uncertainties is described in Sect. 2.6.

## 2.1 From conceptual to process-detailed models of climate tipping elements

In the conceptual network model investigated in this study, the main dynamics of each of the tipping elements are condensed to a non-linear differential equation with two stable states representing the current (baseline) state and a possible transitioned state capturing the qualitative dynamics of generalised tipping elements (see Eq. 1). This serves as a stylised representation for the Greenland Ice Sheet, the West Antarctic Ice Sheet, the AMOC and the Amazon rainforest. We here focus on these four out of a larger range of tipping elements in the cryosphere, biosphere and oceanic and atmospheric circulation patterns that have been suggested in the literature (Schellnhuber et al., 2016; Scheffer et al., 2009; Lenton et al., 2008). In this study, we do not consider a possible "backtipping" (or hysteresis behaviour) of climate tipping elements, since the forcing represented by global mean temperature anomalies is only increased, but never decreased in our experiments. It is clear that the representation of a complex climate tipping element with all its interacting processes as well as positive and negative feedbacks in a stylised cusp bifurcation model is a strong simplification. In the following, we elaborate on why such a cusp bifurcation structure (Eq. 1) can nonetheless be assumed to capture the overall stability behaviour for these four tipping elements (Bathiany et al., 2016), before we introduce more mathematical details of our dynamical systems approach in Sect. 2.3.

1. *AMOC*: Early conceptual models introduced in the 1960s showed that the AMOC could exhibit a cusp-like behaviour, using simplified box models based on the so-called salt-advection feedback (Stommel, 1961; Cessi, 1994). Many extensions and updates to this well-known box model approach have been put forward, each confirming the potential multi-stability of the AMOC (e.g. Wood et al. (2019)). More complex Earth system models including EMICs (e.g., CLIMBER) and AOGCMs (e.g., the FAMOUS and HadGEM3 models) have shown hysteresis behaviour which is qualitatively similar to Eq. 1 (Mecking et al., 2016; Hawkins et al., 2011; Rahmstorf et al., 2005). Furthermore, paleoclimatic evidence suggests a bistability of the AMOC: In paleoclimate records, Dansgaard-Oeschger events (see e.g. Crucifix, 2012) have been associated with large reorganisations of the AMOC (Ditlevsen et al., 2005; Timmermann et al., 2003; Ganopolski & Rahmstorf, 2002), where ice core data links the events to sea-surface temperature increases in the North Atlantic. Even though there are considerable uncertainties, literature estimates suggest the level of global warming sufficient for tipping the AMOC between 3.5–6.0 °C (Schellnhuber et al., 2016; Lenton, 2012; Levermann et al., 2012; Lenton et al.,

2008), with the risk of crossing a critical threshold considerably increasing beyond 4 °C above pre-industrial temperature levels (Kriegler et al., 2009).

2. *Greenland Ice Sheet*: Previous studies have shown that a double fold-like bifurcation structure for the ice sheets can arise from the melt-elevation feedback (Levermann & Winkelmann, 2016) as well as from the Marine Ice Sheet Instability and other positive feedback mechanisms (e.g., DeConto & Pollard, 2016; Schoof, 2007). In particular, dynamic ice sheet model simulations have identified irreversible ice loss once a critical temperature threshold is crossed (Toniazzo et al., 2004), leading to multiple stable states and hysteresis behaviour for the Greenland Ice Sheet (Robinson et al., 2012; Ridley et al., 2010). In Robinson et al. (2012), the critical temperature range for an irreversible disintegration of the Greenland Ice Sheet has been estimated between 0.8–3.2 °C of warming above pre-industrial global mean temperature levels. Paleoclimate evidence further suggests that there have been substantial, potentially self-sustained retreats of the Greenland Ice Sheet in the past. It has, for instance, been simulated that the Greenland Ice Sheet can disintegrate in case warmer ocean conditions from the Pliocene are applied to an initially glaciated Greenland (Koenig et al., 2014). Further, Greenland was nearly ice-free for extended interglacial periods during the Pleistocene (Schaefer et al., 2016). Sea-level reconstructions further suggest that large parts of Greenland could have been ice-free during Marine Isotope Stage 11 and the Pliocene (Dutton et al., 2015).

3. *West Antarctic Ice Sheet*: Compared to the case of the Greenland Ice Sheet, different processes make the West Antarctic Ice Sheet susceptible to tipping dynamics. Since large parts of West Antarctica are grounded in marine basins, changes in the ocean are key in driving the evolution of the ice sheet. The Marine Ice Sheet Instability can trigger self-sustained ice loss where the ice sheet is resting below sea-level on retrograde sloping bedrock (Weertman, 1974; Schoof, 2007). This destabilising mechanism is possibly already underway in the Amundsen Sea region (Favier et al., 2014; Joughin et al., 2014). Once triggered, a single local perturbation via increased sub-shelf melting in the Amundsen region could lead to wide-spread retreat of the West Antarctic Ice Sheet (Feldmann & Levermann, 2015). Further, a recent study shows strong hysteresis behaviour for the whole Antarctic Ice Sheet, identifying two major thresholds which lead to a destabilisation of West Antarctica around 2°C of global warming, and large parts of East Antarctica between 6–9°C of global warming (Garbe et al., 2020). It is likely that the West Antarctic Ice Sheet has experienced brief but dramatic retreats during the past five million years (Pollard & DeConto, 2009). Prior collapses have been suggested from deep-sea-core isotopes and sea-level records (Gasson, 2016; Dutton et al., 2015; Pollard & DeConto, 2005).

4. *Amazon rainforest*: Conceptual models of the Amazon identified multi-stability between rainforest, savannah and treeless states, leading to hysteresis (Staal et al., 2016, 2015; Van Nes et al., 2014). This hysteresis has been found to be shaped by local-scale tipping points of the Amazon rainforest and its resilience might be diminished under climate change until the end of the 21st century (Staal et al., 2020). More complex dynamic vegetation models also found alternative stable states of the Amazon ecosystem (Oyama & Nobre, 2003) and suggest that rainforest dieback might be possible due to drying of the Amazon basin under future climate change scenarios (Nobre et al., 2016; Cox et al., 2004, 2000). Observational data further supports the potential for multi-stability of the Amazon rainforest (Ciemer et al., 2019; Hirota et al., 2011;

Staver et al., 2011). While it remains an open question whether the Amazon has a single system-wide tipping point, the projected increase in droughts and fires (Malhi et al., 2009; Cox et al., 2008) is likely to impact the forest cover on a local to regional scale, which might spread to other parts of the region via moisture-recycling feedbacks (Zemp et al., 2017, 2014; Aragão, 2012). It is important to note that in contrast to the ice sheets and ocean circulation, the rainforest is able to adapt to changing climate conditions to a certain extent (Sakschewski et al., 2016). However, this adaptive capacity

might still be outpaced if climate change progresses too rapidly (Wunderling et al., 2020c). A dieback of the Amazon rainforest has been found under a business-as-usual emissions scenario (Cox et al., 2004), which would be equivalent to a global warming of more than 3 °C above pre-industrial levels until 2100 ($\approx$3.5–4.5 °C (see also Schellnhuber et al., 2016)), mainly due to more persistent El-Niño conditions (Betts et al., 2004).

## 2.2   Physical interpretation of tipping element interactions

Based on these conceptual models as well as building on first coupled experiments with a discrete state Boolean model (Gaucherel & Moron, 2017) and economic impact studies (Cai et al., 2016; Lemoine & Traeger, 2016; Cai et al., 2015), we here describe the interactions of the four tipping elements in a network approach using a set of linearly coupled, topologically equivalent differential equations (Kuznetsov, 2004). In the following we go through the different main interactions of the four tipping elements considered here and expand on the underlying physical processes. Overall, the additional literature

supports and refines the results from an early expert elicitation (Kriegler et al., 2009).

     1. *Greenland Ice Sheet → AMOC*: Increasing freshwater input from enhanced melting of the Greenland Ice Sheet can lead to a weakening of the AMOC, as supported by observations, paleoclimate evidence as well as modelling studies (Caesar et al., 2018; Robson et al., 2014; Driesschaert et al., 2007; Jungclaus et al., 2006; Rahmstorf et al., 2005). Between 1992 and 2018, the Greenland Ice Sheet has lost around 3900$\pm$342 Gt of ice (Shepherd et al., 2020). This ice loss has strongly

accelerated in recent years (Sasgen et al., 2020), and Greenland has been subject to several extreme melt events in the past decade alone (Tedesco & Fettweis, 2020; Nghiem et al., 2012; Tedesco et al., 2011). At the same time, an AMOC weakening of 15% (3$\pm$1 Sv) has been observed since the 1950s (Caesar et al., 2018). This weakening has at least partially been attributed to freshwater influx into the North Atlantic deep water formation regions due to enhanced melting from Greenland. Paleoclimatic records further suggest that the AMOC could exist in multiple stable states, based on observed

temperature changes associated with meltwater influx into the North Atlantic (Blunier and Brook, 2001; Dansgaard et al., 1993). Therefore, it is likely that a tipping of the Greenland Ice Sheet would lead to a destabilisation of the AMOC (see Fig. 1).

     2. *AMOC → Greenland Ice Sheet*: Conversely, if the AMOC weakens, leading to a decline in its northward surface heat transport, Greenland might experience cooler temperatures (e.g. Jackson et al., 2015; Timmermann et al., 2007; Stouffer

et al., 2006), which would have a stabilising effect on the ice sheet. With the global climate model HadGEM3, it has been shown that temperatures in Europe could drop by several degrees if the AMOC collapses, regionally up to 8 °C (Jackson et al., 2015). A cooling trend in sea surface temperatures (SST) over the subpolar gyre, as a result of a weakening AMOC,

has been confirmed by recent reanalysis and observation data (Caesar et al., 2018; Jackson et al., 2016; Frajka-Williams, 2015; Robson et al., 2014). This "fingerprint" translates a reduction in overturning strength by 1.7 Sv per century to 0.44 K SST-cooling per century (Caesar et al., 2018). AMOC regime shifts between weaker and stronger overturning strength during the last glacial period have been associated with large regional temperature changes in Greenland, for example during Dansgaard-Oeschger or Heinrich events (Barker and Knorr, 2016). Moreover, there is paleoclimatic evidence from 3.6 million years ago that a weaker North Atlantic current as part of the AMOC fostered Arctic sea-ice growth which might have preceded continental glaciation in the northern hemisphere at that time (Karas et al., 2020). Based on these findings, we assume that a weakening of the AMOC would have a stabilising effect on the Greenland Ice Sheet (see Fig. 1).

3. *West Antarctic Ice Sheet → AMOC*: It remains unclear whether increased ice loss from the West Antarctic Ice Sheet has a stabilising or destabilising effect on the AMOC (see Fig. 1). Swingedouw et al. (2009) identified different processes based on freshwater hosing experiments into the Southern Ocean, which could be associated with a melting West Antarctic Ice Sheet (Swingedouw et al., 2009). Using the EMIC LOVECLIM1.1, the authors found both enhancing and weakening effects on the AMOC strength:

   First, deep water adjustments are observed. This means that an increase of the North Atlantic Deep Water formation is observed in response to a decrease in Antarctic bottom water production due to the conducted hosing experiment. This mechanism has been termed the so-called bipolar ocean seesaw. Second, salinity anomalies in the Southern Ocean are distributed to the North Atlantic, which dampens the North Atlantic Deep Water formation (compare to Seidov et al., 2005). Third, the North Atlantic Deep Water formation is enhanced by a strengthening of southern hemispheric winds in response to a southern hemispheric cooling. The reason for the stronger winds is the greater meridional temperature gradient between a cooler Antarctic region (due to the hosing experiment) and the equator. This effect has been termed the *Drake Passage effect* (Toggweiler & Samuels, 1995).

   Overall, the first and the third mechanism tend to strengthen the AMOC, while the second process would rather lead to a weakening of the AMOC. The specific time scales and relative strengths of these mechanisms are as of yet unclear (Swingedouw et al., 2009). In a coupled ocean-atmosphere model, a slight weakening of the AMOC was detected for a freshwater input of 1.0 Sv in the Southern Ocean over 100 years (Seidov et al., 2005). However, other studies suggest a stabilisation of the AMOC if influenced by freshwater input from the West Antarctic Ice Sheet due to the effects from the bipolar ocean seesaw by decreasing Antarctic Bottom Water formation as described above (Swingedouw et al., 2008).

4. *AMOC → West Antarctic Ice Sheet*: The interaction from the AMOC to the West Antarctic Ice Sheet is destabilising (see Fig. 1). In case the AMOC shuts down, sea surface temperature anomalies could appear since the northward heat transport is diminished significantly. This could then lead to a warmer south and colder north, as observed in modelling studies (Weijer et al., 2019; Timmermann et al., 2007; Stouffer et al., 2006; Vellinga & Wood, 2002). A model intercomparison study for EMICs and AOGCMs found a sharp decrease of surface air temperatures over the northern hemisphere,

while a slight increase over the southern hemisphere and around the Antarctic Ice Sheet has been observed (Stouffer et al., 2006). In their study (Stouffer et al., 2006), a forcing of 1.0 Sv was applied to the northern part of the North Atlantic Ocean. Therefore, we set this link as destabilising in the interaction network mdoel (see Fig. 1).

5. *Greenland Ice Sheet $\leftrightarrow$ West Antarctic Ice Sheet*: The direct interaction between the Greenland and the West Antarctic Ice Sheet via sea level changes can be regarded as mutually destabilising, however with different magnitudes (see Fig. 1). It is a well-known phenomenon from tidal changes that grounding lines of ice sheets are varying (e.g. Sayag & Worster, 2013). Therefore, the Greenland Ice Sheet and the West Antarctic Ice Sheet could influence each other by sea level rise if one or the other cryosphere element would melt. Gravitational, but also elastic and rotational impacts would then

enhance the sea level rise in case one of the huge ice sheets would melt first, since then only the other ice sheets exerts strong gravitational forces on ocean waters (Kopp et al., 2010; Mitrovica et al., 2009). The impact of this effect would be larger if Greenland becomes ice-free earlier than West Antarctica, since many marine terminating ice shelves are located in West Antarctica, but the interaction is destabilising in both directions (see Fig. 1).

    6. *AMOC $\rightarrow$ Amazon rainforest*: Lastly, the interaction between the AMOC and the Amazon rainforest is set as unclear (see

Fig. 1). It is suspected that the intertropical convergence zone (ITCZ) would be shifted southward in case the AMOC collapses. This could cause large changes in seasonal precipitation on a local scale, and could as such have strong impacts on the Amazon rainforest (Jackson et al., 2015; Parsons, 2014). In the Earth system model ESM2M, it has been found that a strongly suppressed AMOC, through a 1.0 Sv freshwater forcing, leads to drying over many regions of the Amazon rainforest (Parsons, 2014). However, some regions would receive more rainfall than before. On a seasonal level, the wet

season precipitation is diminished strongly, while the dry season precipitation is significantly increased (Jackson et al., 2015; Parsons, 2014). This could have consequences for the current vegetation that is adapted to this partially strong seasonal precipitation. But overall, it remains unclear whether the influence from a tipped AMOC on the precipitation in South America has a reducing or increasing influence. Instead, it might differ from locality to locality and is set as unclear in our study (see Fig. 1).

## 2.3    Dynamic network model of interacting tipping elements

In this subsection, we describe the details of the employed dynamic network model, the foundations of which are given by Eqs. 1 and 2. The critical parameter $c_i$ of tipping element $i$ is modelled as a function of global mean temperature, i.e., $c_i = \sqrt{\frac{4}{27}} \cdot \frac{\Delta \text{GMT}}{T_{\text{limit, i}}}$, where $T_{\text{limit, i}}$ is the critical temperature and $\Delta \text{GMT}$ the increase of the global mean temperature above pre-industrial levels. This parameterization implies that a state change is initiated as soon as the increase of GMT exceeds the

critical temperature ($\frac{\Delta \text{GMT}}{T_{\text{limit, i}}} > 1$, see Table 1). In addition, we model the physical interactions between the tipping elements as a linear coupling (first order approach). The coupling term $\frac{1}{2} \sum_j d_{ij} (x_j + 1)$ consists of a sum of linear couplings to other elements $x_j$ with $d_{ij} = d \cdot s_{ij}/5$. It is necessary to add $+1$ to $x_j$ such that the direction (sign) of coupling is only determined

by $d_{ij}$ and not by the state $x_j$. Thus, Eq. 2 becomes

$$\frac{dx_i}{dt} = \left[ -x_i^3 + x_i + \sqrt{\frac{4}{27}} \cdot \frac{\Delta \text{GMT}}{T_{\text{limit, i}}} + d \cdot \sum_{\substack{j \\ j \neq i}} \frac{s_{ij}}{10} \left( x_j + 1 \right) \right] \frac{1}{\tau_i}. \tag{3}$$

Here $d$ is the overall *interaction strength* parameter that we vary in our simulations and $s_{ij}$ is the link strength based on the expert elicitation (Kriegler et al., 2009) (see Table 2 & Sect. 2.6). The prefactor 1/10 sets the coupling term of Eq. 3 to the same scale as the individual dynamics term by normalising $s_{ij}$ when $d$ is varied between 0.0 to 1.0. The geophysical processes behind the interactions between the tipping elements are listed in Table 2 and are described and referenced in Sect. 2.2.

| Tipping element | $\Delta T_{\text{limit}}$ (°C) |
|---|---|
| Greenland | 0.8 – 3.2 |
| West Antarctica | 0.8 – 5.5 |
| AMOC | 3.5 – 6.0 |
| Amazon rainforest | 3.5 – 4.5 |

**Table 1.** Nodes in the modelled network of interacting tipping elements. For each tipping element in the network (see Fig. 1), a range of critical temperatures $\Delta T_{\text{limit}}$ is known from literature review (Schellnhuber et al., 2016). Within this temperature range, the tipping element is likely to undergo a qualitative state transition.

In this network of tipping elements, very strong interactions exist, as detailed above. For each tipping element, there are two potential reasons for a state transition, either through the increase of GMT or through the coupling to other tipping elements (Fig. 2(a)).

The overall interaction strength $d$ is described as a dimensionless parameter (see Eq. 3) that is varied over a wide range in our simulations, i.e., for $d \in [0; 1]$, to account for the uncertainties in the actual physical interaction strength between the tipping

elements. This way a range of different scenarios can be investigated. An interaction strength of 0 implies no coupling between the elements such that only the individual dynamics remain. When the interaction strength reaches high values around 1, the coupling term is of the same order of magnitude as the individual dynamics term. In principle, more complex and data- or model-based interaction terms could be developed. However, while some interactions (e.g. between Greenland Ice Sheet & AMOC) have been established with EMICs such as CLIMBER-2 and Loveclim as well as GCMs (Wood et al., 2019; Sterl et

al., 2008; Driesschaert et al., 2007; Jungclaus et al., 2006; Rahmstorf et al., 2005), other interactions are less well understood potentially leading to biased coupling strengths (see also Sect. 2.2). Due to the sparsity of data concerning tipping interactions in the past, it remains challenging to extract the interaction parameters from paleoclimatic evidence. We here therefore attempt to include the full uncertainty ranges concerning the different model parameters and interaction strengths. To this end, we run large ensembles of simulations over long time scales. This is important since the disintegration of the ice sheets for instance

| Interaction link | Maximum link strength $s_{ij}$ (a.u.) | Physical process |
|---|---|---|
| Greenland → AMOC | +10 | Freshwater influx |
| AMOC → Greenland | −10 | Reduction of northward heat transport |
| Greenland → West Antarctica | +10 | Sea-level rise |
| AMOC → Amazon rainforest | ±2 up to ±4 | Changes in precipitation patterns |
| West Antarctica → AMOC | ±3 | Increase in meridional salinity gradient (−), |
| | | Fast advection of freshwater anomaly |
| | | to North Atlantic (+) |
| West Antarctica → Greenland | +2 | Sea-level rise |
| AMOC → West Antarctica | +1.5 | Heat accumulation in Southern Ocean |

**Table 2.** Interaction links in the network of tipping elements. For each link in the network of Fig. 1, there is a strength and a sign for each interaction of the tipping elements. The sign indicates if the interaction between the tipping elements is increasing or decreasing the danger of tipping cascades. Following Kriegler et al. (2009), the strength $s_{ij}$ gives an estimate in terms of increased or decreased probability of cascading transitions (Kriegler et al., 2009). E.g., if Greenland transgresses its threshold, the probability that the AMOC does as well is increased by a factor of 10 (see entry for Greenland → AMOC). Then a random number between +1 and $s_{\text{Greenland}\rightarrow\text{AMOC}} = +10$ is drawn for our simulations and used for $s_{ij}$ in Eq. 3. The other way round, the probability that Greenland transgresses its threshold in case the AMOC is in the transitioned state is decreased by a factor of $\frac{1}{10}$. Then a random number between −1 and $s_{\text{AMOC}\rightarrow\text{Greenland}} = -10$ is drawn. The main physical processes that connect pairs tipping elements are described in this table and in Sect. 2.2. The link strengths are grouped into strong, intermediate and weak links. Note that in the expert elicitation (Kriegler et al., 2009), there has been an estimation of the maximum increase or decrease of the tipping probability in case the element which starts the interaction is already in the transitioned state. For example, the link between Greenland and AMOC is given as [1; 10] in Kriegler et al. (2009) and is here modelled as a randomly drawn variable between 1 and 10 for $s_{ij}$. An example for an unclear coupling would be the link between West Antarctica and AMOC which is given as [0.3; 3] in Kriegler et al. (2009) which we translate into an $s_{ij}$ between −3 and 3. In general, the values are drawn between 1 and the respective maximum value $s_{ij}$ if the interaction between $i$ and $j$ is positive or between −1 and the negative maximum value $s_{ij}$ if the interaction between $i$ and $j$ is negative.

would play out over thousands of years (Winkelmann et al., 2015; Robinson et al., 2012). Due to computational constraints, studying such an ensemble of millennial-scale simulations is typically not feasible with more complex Earth System Models. We propagate the considerable uncertainties linked to the parameters of the tipping elements and their interactions with a large-scale Monte-Carlo approach (see Sect. 2.6).

## 2.4   Parameterisation of the tipping elements' intrinsic time scales

The four tipping elements in the coupled system of differential equations form a so-called *fast-slow system* (Kuehn, 2011) describing a dynamical system with slowly varying parameters compared to fast changing states $x_i$. We include the typical transition times $\tau_i$ from the baseline to the transitioned state in Eq. 3 based on literature values (Lenton et al., 2008; Robinson et al., 2012; Winkelmann et al., 2015), setting the tipping time scales for the Greenland Ice Sheet, West Antarctic Ice Sheet, AMOC and the Amazon rainforest to 4900, 2400, 300 and 50 years for a reference warming of 4 °C above pre-industrial GMT,
respectively. The tipping time scale is calibrated at this reference temperature in the case of vanishing interaction between the elements. After calibration, the tipping time is allowed to scale freely with changes in the GMT and the interaction strength $d$. We integrate all model simulations to equilibrium, such that the simulation time is at least 20 times larger than the longest assumed tipping timescale of 4900 years. Since the actual *absolute* tipping times derived from our model simulations are difficult to interpret, our results should not be taken as a projection of how long potential tipping cascades would take to unfold.
Rather, following our conceptual approach, we are interested in the relative differences (not the absolute values) between the typical tipping times as they can be decisive as to whether a cascade emerges or not. Therefore, the figures below show model years in arbitrary units (see Figs. 2 and 3).

## 2.5   Modelling protocol and evaluation of tipping cascades

In our network model, if the critical temperature threshold of a tipping element is surpassed, it transgresses into the transitioned
state (Fig. 2(a)) and can potentially increase the likelihood of further tipping events via its interactions: for instance, the increased freshwater influx from a disintegration of the Greenland Ice Sheet can induce a weakening or even collapse of the AMOC (Fig. 2(b)). In our model simulations, we consider increases of the global mean temperature from 0 up to 8 °C above the pre-industrial average, which could be reached in worst-case scenarios as the extended representative concentration pathway 8.5 (RCP 8.5) by year 2500 (Schellnhuber et al., 2016; IPCC, 2014).
For each tipping element, we start from the *baseline* (non-tipped) state (where $x_i$ is negative). Global warming or interactions with the other parts of the climate system can then cause the element to tip into the *transitioned* state (see Fig. 2). When the critical parameter reaches $\sqrt{\frac{4}{27}}$ from below (i.e., when $\Delta$GMT reaches $T_{\text{limit, i}}$), the stable baseline state $x_i$ reaches $-\frac{1}{\sqrt{3}}$ in case of an autonomous tipping element. Therefore, the threshold for the baseline state is defined as $x_i^- = -\frac{1}{\sqrt{3}}$. If the critical parameter increases above $\sqrt{\frac{4}{27}}$, the state $x_i$ is larger than $x_i^-$, the stability of the lower stable state is lost and a state transition
towards the upper stable $x_i^+$ occurs. Correspondingly to the lower stable state $x_i^-$, the stable transitioned state is defined for states $x_i > x_i^+ = +\frac{1}{\sqrt{3}}$.

We identify and define tipping cascades at a fixed interaction strength $d$ and GMT as the number of additionally tipped elements in equilibrium (as defined above) after an incremental GMT increase of 0.1 °C. The tipping element with critical temperature threshold closest to the GMT at this point, is counted as the initiator of the cascade. All tipping elements that appear in a

particular cascade are counted as an occurring tipping element in that tipping cascade.

With increasing global mean temperature and interaction strength, generally more tipping cascades occur (Fig. 3). However, the size, the timing and the occurrence of cascades can also depend critically on the specific initial conditions (Wunderling et al., 2020b), which are not varied in the experiments presented here. In an exemplary simulation, we show how in one realisation of our Monte Carlo ensemble at low interaction strength, a global mean temperature increase from 1.5°C to 1.6°C triggers the

Greenland Ice Sheet to transition to an ice-free state (Fig. 3(a)). For larger interactions strengths, the West Antarctic Ice Sheet as well as AMOC might then also tip as part of a tipping cascade that was initiated by the Greenland Ice Sheet in this case (Fig. 3(b-c)). The initial conditions and parameters for the specific example of Fig. 3 can be found in supplementary Table S1.

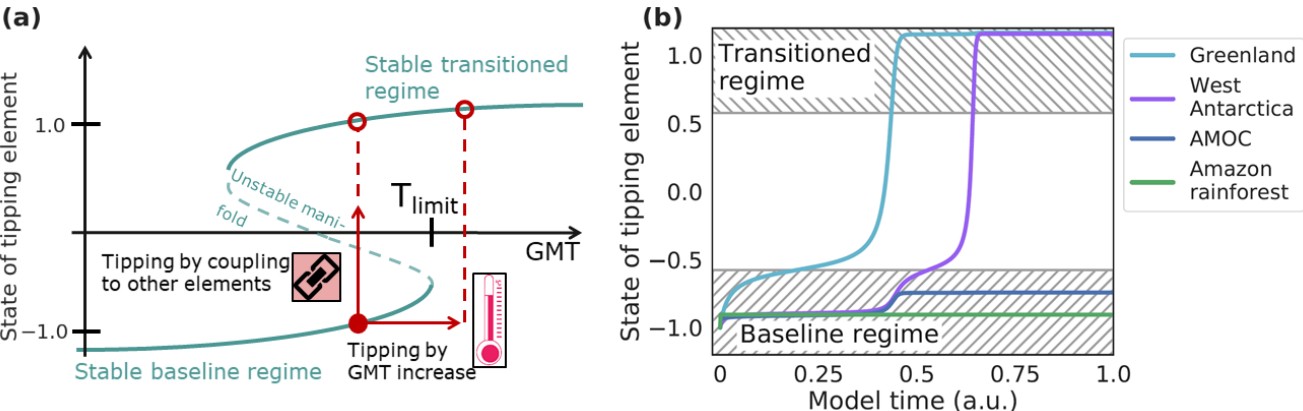

**Figure 2.** Schematic overview of the generalised tipping element and time-series of a tipping cascade. **(a)** Exemplary bifurcation diagram of a tipping element with two stable regimes: The lower state indicates the stable baseline regime, the upper state the stable transitioned regime. In case of the Greenland Ice Sheet, for instance, these correspond to its pre-industrial, almost completely ice-covered state (stable baseline regime) and an almost ice-free state (stable transitioned regime), as can be expected on the long-term for higher warming scenarios (Robinson et al., 2012). There are two ways how a tipping element can transgress its critical threshold (*unstable manifold*) and move into the transitioned state, either by an increase of global mean temperature or via interactions with other climate components. In both cases, the tipping element converges to the stable transitioned regime indicated by the red hollow circles. **(b)** Exemplary time series showing a tipping cascade of two elements. Here, Greenland transgresses its critical temperature ($T_{\text{limit, Greenland}}$) first, i.e., would become ice-free. Through its interaction with the West Antarctic Ice Sheet, the West Antarctic Ice Sheet then transgresses the unstable manifold in vertical direction (following the path of the red upward directed arrow in panel **(a)**). This example is based on a scenario with global mean temperature increase of 1.6 °C above pre-industrial levels and an interaction strength $d = 0.16$ (see also Fig. 3).

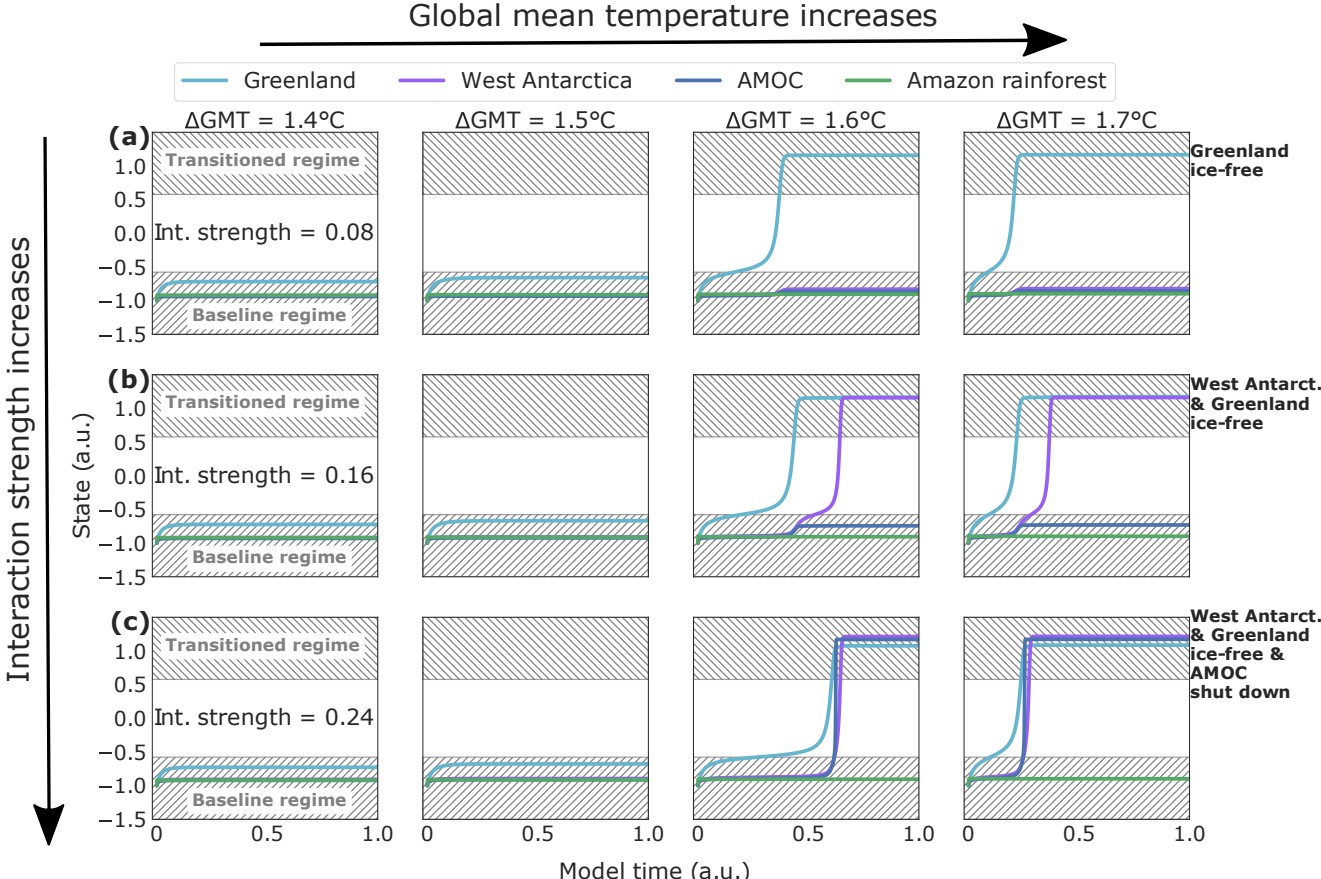

**Figure 3.** Time series of tipping cascades. Exemplary time series of states for each of the four investigated tipping elements, here simulated until equilibrium is reached. For comparability reasons, the parameter settings for the time series are the same (exact parameters can be found in Table S1) and all time series are computed for ΔGMT increases of 1.4, 1.5, 1.6 and 1.7 °C above pre-industrial (columns). Couplings are constant for each row. Tipping cascades as shown here are defined as the number of transitioned elements at a fixed interaction strength and ΔGMT compared to the simulation with a slightly higher ΔGMT (ΔGMT increase by 0.1 °C), but same interaction strength. If, between these two simulations, some of the tipping elements alter their equilibrium state, then a tipping cascade of the respective size occurred and is counted as such. **(a)** Singular tipping event for an interaction strength of 0.08. Tipping occurs at 1.6 °C. **(b)** Tipping cascade of size two for an interaction strength of 0.16. **(c)** Tipping cascade of size three for an interaction strength of 0.24. For other initial conditions, interaction strengths and global mean temperatures (ΔGMT) tipping cascades of size four can occur, too. Additionally, we marked the baseline and the transitioned regime as grey hatched areas. Between the hatched areas, the state is not stable and a critical state transition occurs. In the lower grey area, the element is called to be in the *baseline* regime and in the *transitioned* regime in the upper grey region.

## 2.6 Monte Carlo sampling and propagation of uncertainties

Since the strength of interactions between the tipping elements is highly uncertain, a dimensionless interaction strength is varied over a wide range in our network approach to cover a multitude of possible scenarios. To cope with the uncertainties in the critical threshold temperatures and in the link strengths between pairs of tipping elements (see Eq. 3, Tables 1 and 2), we set up a Monte-Carlo ensemble with approximately 3.7 million members in total.

This Monte Carlo ensemble is generated as follows: for each combination of global mean temperature $\Delta$GMT and overall
interaction strength $d$, we create 100 realisations of randomly drawn parameter sets for critical threshold temperatures $T_{\text{limit},i}$ and interaction link strengths $s_{ij}$ based on the uncertainty ranges given above (see Tables 1 and 2). Since our model has 11 parameters with uncertainties (4 critical threshold temperature parameters and 7 interaction link strength parameters), we use a latin-hypercube sampling to construct a set of parameters for each ensemble members such that the multidimensional space of sampled parameters is covered better than with a usual random sample generation (Baudin, 2013).

We also sample all 9 different interaction network structures which arise when we permute all possibilities (negative, zero, positive) arising from the two unclear links between AMOC and Amazon rainforest, and between West Antarctica and the AMOC (see Table 2 and Fig. 1). For each of these 9 network structures, we compute the same 100 starting conditions that we received from our latin-hypercube sampling. Thus, in total, we compute 900 samples for each GMT ($0.0 - 8.0$ °C, step width: $0.1$ °C) and interaction strength ($0.0 - 1.0$, step width: $0.02$) combination resulting in a large ensemble of 3.7 million members
overall.

Our approach is conservative in the sense that there are several destabilising interactions which are not considered here (Lenton et al., 2019; Steffen et al., 2018). Further, by sampling uncertain parameters from a uniform distribution, we are treating lower and higher threshold temperatures as well as strong and weak link interactions equally, potentially resulting in a more balanced ensemble. Additional knowledge about the critical threshold temperatures and interaction link strengths would considerably
improve our analysis.

## 3 Results

### 3.1 Shift in effective critical threshold temperatures due to interactions

Owing to the interactions between the tipping elements, their respective critical temperatures (previously identified for each
element individually, see Fig. 4(a)) are effectively shifted to lower values (except for Greenland, see Figs. 4(b) and (c)). For West Antarctica and the AMOC, we find a sharp decline for interaction strengths up to 0.2 and an approximately constant critical temperature range afterwards. The effective critical temperature for the Amazon is only marginally reduced due to the interactions within the network, since it is only influenced by the AMOC via an unclear link.

In particular, the ensemble average of the critical temperature at an interaction strength of $d = 1.0$ is lowered by about 1.2 °C

(≈40%) for the West Antarctic Ice Sheet, 2.75 °C (≈55%) for the AMOC and 0.5 °C (≈10%) for the Amazon rainforest, respectively (see Fig. S2). This is likely due to the predominantly positive links between these tipping elements (see Fig. 1). In contrast, the critical temperature range for the Greenland Ice Sheet tends in fact to be raised due to the interaction with the other tipping elements, accompanied by a significant increase in overall uncertainty. This can be explained by the strong negative feedback loop between Greenland and the AMOC that is embedded in the assumed interaction network (see Table 2,

see also Gaucherel & Moron (2017)). On the one hand, enhanced meltwater influx into the North Atlantic might dampen the AMOC (positive interaction link), while on the other hand, a weakened overturning circulation would lead to a net-cooling effect around Greenland (negative interaction link). Thus, the state of Greenland strongly depends on the specific parameter values in critical threshold temperature and interaction link strength of the respective Monte-Carlo ensemble members. Overall, the interactions are more likely to lead to a destabilisation within the network of climate tipping elements with the

exception of the Greenland Ice Sheet.

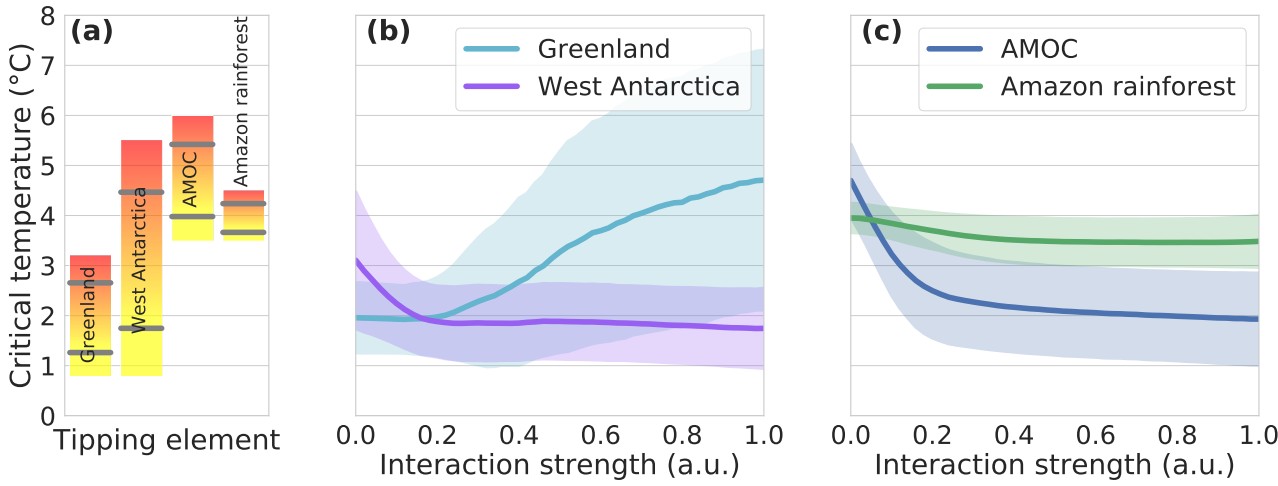

**Figure 4.** Shift of critical temperature ranges due to interactions. **(a)** Critical global mean temperatures for each of the four investigated tipping elements without taking interactions into account (as reproduced from literature; Schellnhuber et al. (2016)). Grey bars indicate the standard deviation arising when drawing from a random uniform distribution between the respective upper and lower temperature limits. These bars correspond to the critical temperature ranges in case of zero interaction strength in panels **(b)** and **(c)**. **(b, c)** Change of critical temperature ranges with increasing interaction strength for the Greenland Ice Sheet and West Antarctic Ice Sheet (panel **(b)**) and the Atlantic Meridional Overturning Circulation (AMOC) and the Amazon rainforest (panel **(c)**). The standard deviation of the critical temperatures for each tipping element within the Monte Carlo ensemble is given as respective colour shading.

## 3.2 Risk of emerging tipping cascades

Tipping cascades occur when two or more tipping elements transgress their critical thresholds for a given temperature level (see Sect. 2.5). We evaluate the associated risk as the share of ensemble simulations in which such tipping cascades are detected. For global warming up to 2.0 °C, tipping occurs in 61% of all simulations (Fig. 5(a)). This comprises the tipping of individual elements (22%) as well as cascades including 2 elements (21%), 3 elements (15%) and 4 elements (3%; see Fig. 5(b)). Since the coupling between the tipping elements is highly uncertain, we introduce an upper limit to the maximum interaction strength and vary it from 0.0 to 1.0 (see Table 3). The highest value of 1.0 implies that the interaction between the elements is as important as the nonlinear threshold behaviour of an individual element (see Eq. 3). For lower values, the interaction plays a less dominant role. We find that the occurrence of tipping events does not depend significantly on the maximum interaction strength – however, the cascade size decreases for lower values.

| Maximum interaction strength $d$ | No tipping (%) | Tipping (%) | Cascade sizes (%) | | | |
| --- | --- | --- | --- | --- | --- | --- |
| | | | 1 | 2 | 3 | 4 |
| 1.0 | 39 | 61 | 22 | 21 | 15 | 3 |
| 0.75 | 39 | 61 | 26 | 18 | 14 | 2 |
| 0.50 | 39 | 61 | 31 | 15 | 14 | 1 |
| 0.25 | 39 | 61 | 42 | 13 | 6 | 0 |
| 0.10 | 39 | 61 | 56 | 5 | 0 | 0 |

**Table 3.** Share of tipping events in ensemble simulations. For different maximum values of the interaction strength $d$ (first column), the share of ensemble simulations is shown that have a tipping event or cascade (third column) within the Paris limit until the global mean temperature increase reaches 2.0 °C above pre-industrial. This means that 61% of all ensemble members contain a tipping event or cascade, while 39% do not (second column) if all interaction strengths until 1.0 are considered (see Figs. 5(a, b)). Overall, the fraction of tipping events stays the same and does not decrease for lower maximum interaction strengths. However, the distribution of tipping events and cascade sizes changes, i.e., the number of large cascades decreases with lower maximal interaction strength. This is shown in the split last column that displays the share of cascades of size one, two, three and four.

Tipping cascades are first induced at warming levels around 1 °C above pre-industrial GMT, where the lower bound of the critical temperature range for the Greenland Ice Sheet is exceeded. The bulk of tipping cascades, however, is found between 1 and 3 °C GMT increase. This is true for all cascade sizes (see Fig. 5(c, d, e)). For temperatures above 3 °C GMT increase, cascades occur less frequently since most of the tipping elements already transgress their individual threshold before this temperature is reached. The most prevalent tipping cascades of sizes two and three, as simulated in our network approach, consist of cascading transitions between the ice sheets and/or the AMOC, summing up to 80% of all tipping cascades of sizes two or three (Fig. 5(f)).

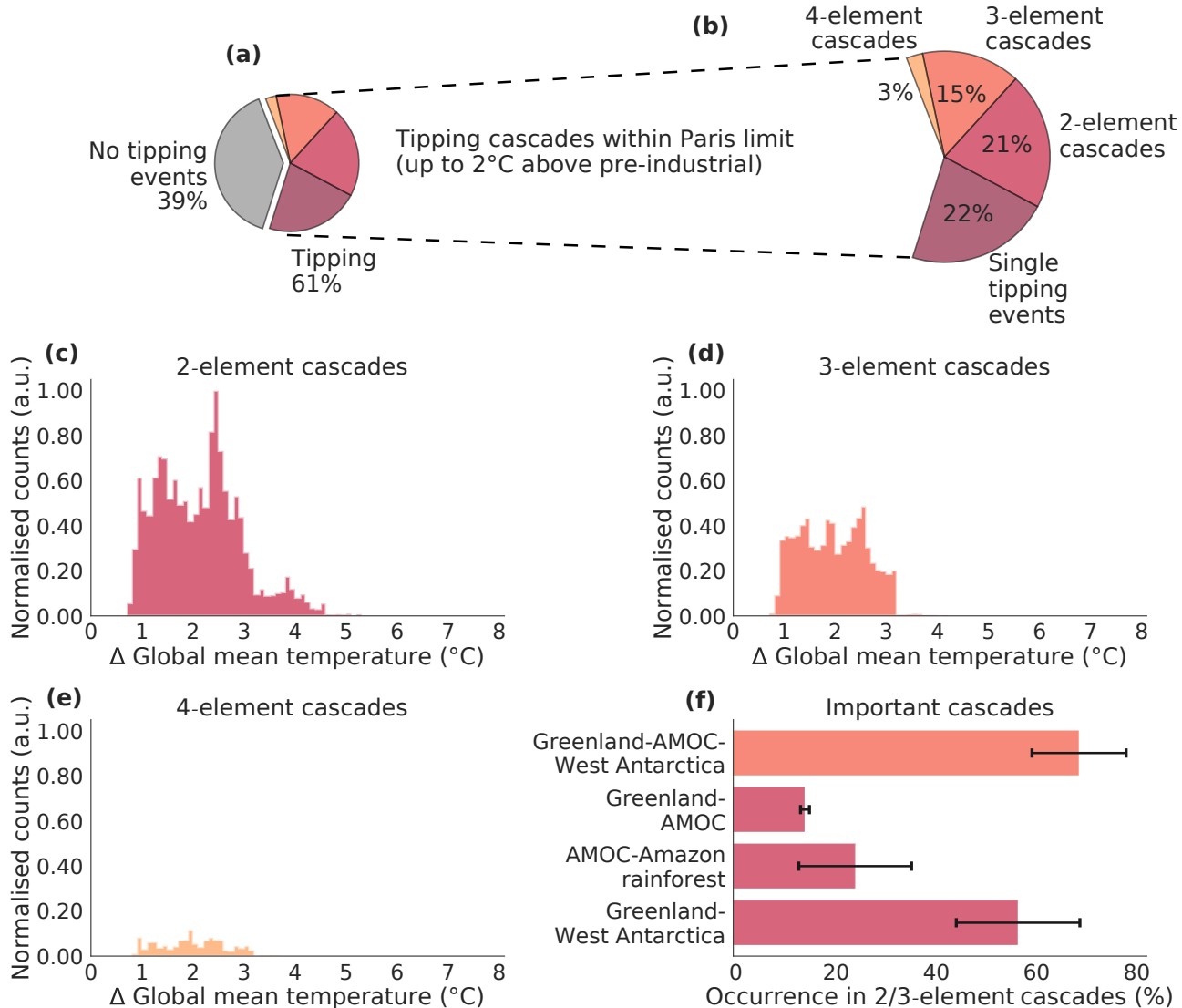

**Figure 5.** Tipping cascades for all interaction strengths between 0.0 and 1.0. **(a, b)** For global warming up to 2.0 °C above pre-industrial levels, the colour shading illustrates the fraction of model representations in the Monte-Carlo ensemble without tipping events (grey), with a singular tipping event (purple) and with cascades including two (red), three (dark orange) and four (light orange) elements. **(c, d, e)** Occurence of tipping cascades of size two, three, and four as a function of global mean temperature increase. The counts are normalised to the highest value of the most frequent tipping cascade (in cascades of size two). **(f)** Dominant cascades of size two and three for temperature increases from $0 - 8$ °C above pre-industrial. Other cascades are not shown, since their relative occurrence is comparatively much smaller. The standard deviation represents the difference between the possible ensemble realisations of the interaction network (see Sect. 2.3). Hence, it tends to be larger for cascades where unclear interaction links are involved, e.g., for the AMOC-Amazon rainforest cascade (compare Fig. 1 and Table 2).

### 3.3 Different roles of tipping elements

For each of the four tipping elements, we systematically assess their role within the network model, generally distinguishing
between initiators (triggering a cascade), followers (last element in a tipping chain) and mediators (elements in-between).

We find that in up to 65% of all ensemble simulations, the Greenland Ice Sheet triggers tipping cascades. At the same time, it occurs as frequently in cascades as the other tipping elements (around 29% of all cases, see Fig. 1). Thus, we call Greenland a *dominant initiator* of cascades. Following this argument for Greenland, the West Antarctic Ice Sheet is both an *initiator and mediator* of cascades, since it occurs often in cascades (31%) and, likewise, often acts as the initiator (23%). Although the
frequency of occurrence in cascades is very similar for the AMOC as for the two large ice sheets, it is a *dominant mediator* of cascades since it does not initiate many cascades (13%). Lastly, the Amazon rainforest is a *pure follower* in cascades because it is only influenced directly by the AMOC and cannot influence any other tipping element itself in our model due to the given interaction network structure (see Fig. 1). The reason why the ice sheets often act as initiators of tipping cascades in our model is likely because their critical threshold ranges tend to be lower than for the other tipping elements (see Fig. 4a). Many
cascades are then passed on to other tipping elements, especially the AMOC. Thus, the role of the AMOC as the main mediator of cascades can be understood from a topological point of view since the AMOC is the most central network element with many connections to the other tipping elements. As such, the AMOC connects the two hemispheres and can be influenced by both the Greenland Ice Sheet and the (West) Antarctic Ice Sheet as is also suggested by literature (Wood et al., 2019; Ivanovic et al., 2018; Hu et al., 2013; Swingedouw et al., 2009; Rahmstorf et al., 2005).

### 425 3.4 Structural robustness and sensitivity analysis including ENSO

While many tipping elements (including the ice sheets, AMOC and Amazon rainforest) to a first approximation exhibit a transition between two or more alternative stable states, often described by the paradigmatic double-fold bifurcation (Scheffer et al., 2009; Lenton et al., 2008) as discussed above, tipping of the El-Niño Southern Oscillation (ENSO) rather could imply a transition from irregular oscillatory occurrences to a more permanent state of strong El-Niño conditions (Dekker et al., 2018;
Lenton et al., 2008; Kriegler et al., 2009). In coupled experiments for AMOC and ENSO with conceptual models, it was found that a changing AMOC could trigger a tipping of ENSO (Dekker et al., 2018; Timmermann et al., 2005). Overall, changes in the frequency of major El-Niño events seem likely, also based on intermediate complexity and conceptual models (Dekker et al., 2018; Timmermann et al., 2005), but whether this poses the possibility of a permanent El-Niño state remains debated. A more frequent occurrence of El-Niño events could have strong impacts on global ecosystems up to a potential dieback of the
Amazon rainforest (Duque-Villegas et al., 2019).

While some studies emphasise the uncertainty about future ENSO changes (Kim et al., 2014; Collins et al., 2010), another study found that the frequency of El-Niño events could increase twofold in climate change scenarios in simulations of the CMIP3 and CMIP5 climate model ensembles as well as in perturbed physics experiments (Cai et al., 2014). Also, some ENSO characteristics appear to respond robustly to global warming (Kim et al., 2014; Power et al., 2013; Santoso et al., 2013), such
as an intensification of ENSO-driven drying in the western Pacific and rainfall increases in the central and eastern equatorial

Pacific due to nonlinear responses to surface warming (Power et al., 2013). Moreover, from an observational point of view, it was found that the global warming trend since the early 1990s has enhanced the Atlantic capacitor effect which might lead to more favourable conditions for major El-Niño events on a biennial rhythm (Wang et al., 2017). Paleoclimate evidence from the Pliocene (5.3–2.6 Myr before present) with atmospheric $CO_2$ levels comparable to today's conditions suggests that there may have been permanent El-Niño conditions during that epoch (Fedorov et al., 2006; Ravelo et al., 2006; Wara et al., 2005). However, it must be noted that the Pliocene was different in terms of the continental configuration compared to today. Particularly, the Panama gateway was open for at least part of the Pliocene resulting in tropical interactions between Atlantic and Pacific ocean waters (Haug & Tiedemann, 1998).

Given the particular uncertainties regarding ENSO compared to the other tipping elements considered in our analysis, we excluded it and its interactions with the other tipping elements in the main analysis above. However, we performed a comprehensive structural robustness and sensitivity analysis including ENSO as a tipping element (see also Supplementary Figs. S3 – S8): For this purpose, we choose to represent ENSO in the same way as the other tipping elements, although the use of Eq. 1 is not entirely appropriate for ENSO. Rather, the potential tipping behaviour could be conceptualised by a Hopf-bifurcation (i.e., a transition from a limit cycle leading to oscillating behaviour to a stable fixed point attractor) instead of a fold bifurcation (Dekker et al., 2018; Timmermann et al., 2003; Zebiak & Cane, 1987).

A typical transition time of 300 years is chosen, the critical temperature threshold lies between 3.5–7.0°C above pre-industrial levels (Schellnhuber et al., 2016) and our analysis is based on simulations of 11 million ensemble members arising from the 27 different network combinations from the three unclear links AMOC → Amazon rainforest, West Antarctica → AMOC and Amazon rainforest → ENSO (see Fig. S3). The interactions including ENSO are described in detail in the Supplement (see Tab. S2 and description there).

Our robustness analysis reveals that the roles of the tipping elements remain qualitatively the same: the ice sheets remain strong initiators of tipping cascades (in 40% of cases for the Greenland Ice Sheet, and 28% of cases for the West Antarctic Ice Sheet). The AMOC mainly acts as a mediator and only initiates 5% of all cascades (see Fig. S3). In this extended network of tipping elements, ENSO tends to take on an intermediate role. Since it is strongly coupled to the Amazon rainforest, it initiates many cascades including the Amazon rainforest, especially at temperature levels above 3 °C (see Fig. S4). But apart from that, ENSO also mediates tipping cascades from the AMOC to the West Antarctic Ice Sheet or the Amazon rainforest. Generally, we also find that the interactions destabilise the overall network of tipping elements apart from the Greenland Ice Sheet (Figs. S5 and S6). The change in the critical temperature range for the Amazon rainforest is larger and is shifted more towards lower temperature levels due to the influence from ENSO. Overall, the model results remain robust, also with respect to the occurrence and size of tipping cascades (see Fig. S7), suggesting a certain degree of structural stability of our analysis.

## 4 Discussion and Conclusions

It has been shown previously that the four integral components of the Earth's climate system mainly considered here are at risk of transgressing into undesirable states when critical thresholds are crossed (Schellnhuber et al., 2016; Lenton et al.,

2008). Over the past decades, significant changes have been observed for the polar ice sheets, as well as for the Atlantic Meridional Overturning Circulation (AMOC) and the Amazon rainforest (Lenton et al., 2019). Should these climate tipping elements eventually cross their respective critical temperature thresholds, this may affect the stability of the entire climate system (Steffen et al., 2018).

In this study, we show that this risk increases significantly when considering interactions between these climate tipping elements and that these interactions tend to have an overall destabilising effect. Altogether, with the exception of the Greenland Ice Sheet, interactions effectively push the critical threshold temperatures to lower warming levels, thus reducing the overall stability of the climate system. The domino-like interactions also foster cascading, nonlinear responses. Under these circumstances, our model indicates that cascades are predominantly initiated by the polar ice sheets and mediated by the AMOC. Therefore, our results also imply that the negative feedback loop connecting the Greenland Ice Sheet and the AMOC might not be able to stabilise the climate system as a whole, a possibility that was raised in earlier work using a boolean modelling approach (Gaucherel & Moron, 2017).

While our conceptual model evidently does not represent the full complexity of the climate system and is not intended to simulate the multitude of biogeophysical processes or to make predictions of any kind, it allows us to systematically assess the qualitative role of known interactions of some of the most critical components of the climate system. The large-scale Monte Carlo approach further enables us to systematically take into account and propagate the substantial uncertainties associated with the interaction strengths, interaction directions and the individual temperature thresholds. This comprehensive assessment indicates structurally robust results that allow qualitative conclusions, despite all these uncertainties.

In our Monte Carlo approach employed for propagating parameter uncertainties, we assume that all parameters including critical threshold temperatures and interaction link strengths are statistically independent. However, this is likely not the case in the climate system where for example interaction link strengths associated with the AMOC to Greenland and West Antarctica would be expected to be correlated. Further analyses would have to consider the effects of such interdependencies.

Overall, this work could form the basis of a more detailed investigation using more process-detailed Earth System Models that can represent the full dynamics of each tipping element and their interactions. Major advances have been made in developing coupled Earth System Models, however, computational constraints have so far prohibited a detailed interaction analysis as is presented in this work. In the future, these more complex climate models might be driven with advanced ensemble methods for representing and propagating various types of uncertainties in climate change simulations (Daron & Stainforth, 2013; Stainforth et al., 2007), which would comprise a significant step forward in the current debate on nonlinear interacting processes in the realm of Earth system resilience. Some examples of relevant processes that could be investigated with more complex models are the following: First, the changing precipitation patterns over Amazonia due to a tipped AMOC, i.e., whether rainfall patterns will increase or decrease and whether this would be sufficient to induce a tipping cascade in (parts of) the Amazon rainforest. This would shed more light on the interaction pair AMOC-Amazon rainforest. Second, the influence of the disintegration of the West Antarctic Ice Sheet on the AMOC could be further studied by introducing freshwater input into the Southern Ocean surrounding the West Antarctic Ice Sheet similar to the hosing experiments that have been performed for the Greenland Ice Sheet (Wood et al., 2019; Hawkins et al., 2011; Rahmstorf et al., 2005). Here, some studies suggest that

freshwater input into the Southern Ocean at a modest rate would not impact the AMOC as much as freshwater input into the North Atlantic (Ivanovic et al., 2018; Hu et al., 2013; Swingedouw et al., 2009), while higher melt rates could have more severe impacts on the AMOC (Swingedouw et al., 2009). With carefully calibrated coupled ice-ocean models, including dynamic ice sheets (e.g. Kreuzer et al., 2020), ice-ocean tipping cascades could be studied in more detail.

Further, the timescales for potential tipping dynamics need to be more rigorously explored in contrast to the conceptual approach used here. It is important to note that the transition of one tipping element has a delayed effect on the other elements, especially in the case of the comparatively slowly evolving ice sheets. Their temperature threshold is lower than for the other tipping elements considered here and their disintegration would unfold over the course of centuries up to millennia (Winkelmann et al., 2015; Robinson et al., 2012; Lenton et al., 2008). Therefore, meltwater influx into the ocean and changes in sea level would affect the state of other tipping elements only after a significant amount of time. Our analysis of emerging tipping cascades therefore needs to be understood in terms of committed impacts over long time scales due to anthropogenic interference with the climate system mainly in the 20th and 21st centuries, rather than short-term projections.

Finally, it appears worthwhile to perform an updated expert elicitation along the lines of Kriegler et al. (2009), where additional interactions, tipping elements and a better understanding of the interaction strengths would help to narrow down the space of possible scenarios and uncertainties that have been investigated here.

*Code and data availability.* The data that support the findings of this study are available from the corresponding author upon reasonable request. The code for the Monte Carlo ensemble construction and the conceptual network model that support the findings of this study are freely (3-clause BSD license) available on github under the following doi: 10.5281/zenodo.4153102.

*Author contributions.* R.W. and J.F.D. conceived the study. R.W., J.F.D. and N.W. designed the model experiments. N.W. conducted the model simulation runs and prepared the figures. All authors discussed the results and wrote the manuscript.

*Competing interests.* The authors declare no competing interests.

*Acknowledgements.* This work has been carried out within the framework of the IRTG 1740/TRP 2015/50122-0 funded by DFG and FAPESP. N.W., J.K. and R.W. acknowledge their support. N.W. is grateful for a scholarship from the Studienstiftung des Deutschen Volkes. J.F.D. is grateful for financial support by the Stordalen Foundation via the Planetary Boundary Research Network (PB.net), the Earth League's EarthDoc program, and the European Research Council Advanced Grant project ERA (Earth Resilience in the Anthropocene; grant ERC-2016-ADG-743080). R.W. acknowledges support by the European Union's Horizon 2020 research and innovation programme under grant agreement no. 820575 (TiPACCs). We are thankful for support by the Leibniz Association project DominoES. The authors gratefully acknowledge the European Regional Development Fund (ERDF), the German Federal Ministry of Education and Research and the Land Bran-

denburg for supporting this project by providing resources on the high performance computer system at the Potsdam Institute for Climate Impact Research. We thank Anders Levermann, Marc Wiedermann, Jobst Heitzig, Niklas Kitzmann and Julius Garbe for fruitful discussions. We are also grateful to Jonathan Krönke for support with the software package "pycascades".

540

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
