# Peer review of "Interacting tipping elements increase risk of climate domino effects under global warming"

_Earth System Dynamics, 2020_

## Referee Comment (RC1) · Anonymous Referee #1 · 6 Apr 2020

The authors are discussing domino effects of individual tipping elements when the elements are coupled. The authors are investigating a set of coupled systems for evaluating how tipping phenomena cascade in the coupled systems. My major concern is that although the authors succeeded in showing how tipping phenomena cascade in the toy model, the set of coupled systems cannot imply anything related to the actual climate, in my opinion, because the set of coupled systems is too conceptual to infer something in a real climate. Thus, the authors should discuss this gap more carefully to clarify the limitations as well as surely ensured implications of the current study. This is my major concern.

There are some minor points as well. 1. Line 6, "between" should be "among". 2. Line 7, insert "(AMOC)" after "Atlantic Meridional Overturning Circulation". 3. Line 35,

"more simple" should be "simpler". 4. Line 86, "more simple" should be "simpler".

---

## Referee Comment (RC2) · Anonymous Referee #2 · 20 Apr 2020

This paper presents an interesting extention of the Dekker et al. (2018) work on cascading tipping behavior by considering the possible cascading interactions of five potential tipping elements. The strenght of the paper is clearly the large-scale Monte Carlo approach such that the overall behavior of the dynamical system (1) is studied. A clear weakness of the paper is the connection to climate dynamics. I suggest that the authors try to rewrite the paper to strengthen the latter aspect; the comments below are intended to help with this.

1. Whereas one could justify (e.g. from conceptual models) that saddle-node bifurcations, underlying the individual dynamics term in (1), are relevant for the AMOC, ice sheets and Amazon rain forest, this does not hold for ENSO. Although this is mentioned in the paper (e.g. l63-64 and l75-78), there is no discussion on this issue. ENSO is also

[Figure]

problematic because its behavior may not change substantially under climate change (e.g., in CMIP5 models (Kim et al., 2014).) The best way out is probably to omit ENSO from the list of tipping elements; the results will very likely still be interesting. If the authors want to keep ENSO, they should better justify the use of (1) for this tipping element.

2. The coupling between the tipping elements is too sketchy at the moment and requires more discussion and analysis. The coupling terms $s_{i,j}$ are now more or less 'guessed' but there are results of EMICs (e.g., Climber, Loveclim and modern variants) where such linear coupling coefficients could be estimated (e.g. from regression analysis). This would also shed more light on the part of the state vector ($x_i$ in (1)) where the coupling occurs (as now only sketched in Table 2). Dekker et al. (2018) have done this to establish the relation between the AMOC and ENSO (meridional Atlantic temperature difference and the equatorial wind stress). I realize that this is more work, but it would enhance the quality and possible impact of the paper significantly.

3. With ENSO removed and a better justification of the linear coupling from EMIC results (points 1 and 2 above), the interpretation of the results in Fig. 4-6 can be much improved. This in particular holds for the interesting result that the coupling destabilizes the reference climate state (as mentioned in the paper l268-270). Section 4 can then be substantially improved and it would be particularly helpful to the community if suggestions would be given on climate model experiments (even with EMICs) which could test the occurrence of this cascading behavior.

There are several more minor issues but as the paper probably is rewritten substantially with different results, I will not mention these here.

---

## Author Comment (AC1) · 14 Jun 2020

The full response to the reviewers' comments can be found below as a supplementary PDF comprising the comments from both reviewers and the new simulations without ENSO as a tipping element as was recommended by the reviewers. Thank you very much for your comments.

Best regards, Nico Wunderling, Jonathan Donges, Jürgen Kurths, Ricarda Winkelmann

Please also note the supplement to this comment:
https://esd.copernicus.org/preprints/esd-2020-18/esd-2020-18-AC1-supplement.pdf

[Figure]

**Supplement:**

Dear Mr Crucifix, Dear Reviewers,

Thank you very much for the provided reviews on our manuscript "Interacting tipping elements increase risk of climate domino effects under global warming". The comments help us to sharpen and improve our research paper substantially.

The main changes in the manuscript in response to the comments from the reviewers comprise:

1. Structural robustness analysis with respect to ENSO (see also Figs. N1 to N4 below)
2. Change framing of introduction to highlight the usefulness of the conceptual approach as a hypothesis generator for more process-detailed studies
3. Change framing of the conclusions of our results to improve the helpfulness to the community for instance by including ideas on which experiments could be used to investigate interactions of tipping elements in more process-based Earth system models in the future.
4. Improved motivation for the choice of interaction functions and interaction strength  parameters.

Please find below a detailed point-by-point response to the reviewer comments. We are happy to include the proposed changes to our manuscript.

Sincerely yours,
Nico Wunderling, Jonathan Donges, Jürgen Kurths & Ricarda Winkelmann

**Reviewer #1:**

The authors are discussing domino effects of individual tipping elements when the elements are coupled. The authors are investigating a set of coupled systems for evaluating how tipping phenomena cascade in the coupled systems. My major concern is that although the authors succeeded in showing how tipping phenomena cascade in the toy model, the set of coupled systems cannot imply anything related to the actual climate, in my opinion, because the set of coupled systems is too conceptual to infer something in a real climate. Thus, the authors should discuss this gap more carefully to clarify the limitations as well as surely ensured implications of the current study. This is my major concern.

**We agree with the reviewer that our model is based on a couple of simplifications that should be made clearer with respect to its limitations. Opposed to a toy model, we view our model as a hypothesis generator with which we can arrive at interesting and valuable qualitative results (*hypotheses*) that can then, afterwards, be examined or checked by more complex models such as EMICs or GCMs. On the other side, it is and cannot be our aim with this model to make pure quantitative statements, let alone predictions.**

**Contrasting this, we see this study as a step towards more in-depth studies using more complex EMICs or GCMs. But despite the many recent progresses in EMICs and GCMs, these models would be required to resolve the nonlinear behavior of all or a sufficient subset of tipping elements which is not yet the case to our knowledge. On top of that, computational constraints might have hindered such a wide analysis as of yet which has also been mentioned for instance in Wood et al. (Clim. Dyn., 2019). These problems are overcome by our, admittedly simplified, model with its merits on the qualitative side (eg. role of tipping elements and impacts of interactions) rather than on exact quantifications or even predictions. This is why we feel that the *conceptual investigation* of the Earth system with emulator-like models such as this is worthwhile and creates interesting insights into the dynamics of interacting tipping elements. We will rewrite the introduction with such a section in more detail.**

**In our manuscript, the main qualitative findings are:**

1. **We find that the ice sheets are the tipping elements which are most likely to initiate cascades of tipping events. Especially tipping cascades from the Greenland Ice Sheet to the AMOC as well as (less, but also significant) from the West Antarctic Ice Sheet to the AMOC fall into this category (see Fig. 1 and Fig. 6e,f of the main manuscript). This is also supported by many literature studies with conceptual models (eg. Wood et al., Clim. Dyn. 2019, Stommel, Tellus 1961, etc.), where the AMOC is influenced by freshwater input, mainly from a melting Greenland Ice Sheet. Furthermore, a reduction of the AMOC has also been found in data and general circulation models (Caesar et al., Nature 2018, Rahmstorf et**

al., Nat. Clim. Change, 2015, Hawkins et al., GRL 2011). Thus we consider this result as very robust and physically meaningful.

2. While the ice sheets are the initiators of tipping cascades, we found that the AMOC is a mediator/transmitter of cascades. In our model, this results merely comes from the fact that the AMOC has the most connections to other tipping elements (i.e., from a topological point of view). But also in the real climate system, the AMOC connects the two polar regions and the equator due to its specific structure influenced by the melting ice from Greenland as well as from Antarctica (eg. Swingedouw et al., Clim. Dyn. 2009, Hu et al., Journal of Climate 2013). Thus, we think that this role as a connector would remain in a more process based study with, for instance, GCMs.

3. Furthermore, we think that the reduction of the critical temperatures for all tipping elements but Greenland is also a robust result that would hold under further studies (see Fig. 4 and Fig. 5).

In a revised version of the manuscript, we would emphasize these points made here and write it in a clearer way, also emphasizing that future research should aim at attaining more details on the specific interaction pairs.

On top of that, we performed a structural robustness analysis and recomputed all our results without ENSO since it is debated whether ENSO is a tipping element and how a tipped state might play out with climate change, e.g.: will ENSO be permanent, will it be stronger, …? Please also have a look into our comment to reviewer #2 on more details about the literature on ENSO under climate change and paleo evidence.
Thus, to improve our model, we recomputed all our results to check for robustness without taking ENSO into account as a tipping element, see new Figs N1 to N4.

The main messages regarding the role of the tipping elements still hold and, thus, are robust (the ice sheets (mainly Greenland) are initiators, the AMOC transmits cascades, see Fig. N3). Also, the critical temperature for West Antarctica and AMOC goes down, while the critical temperature for the Greenland Ice Sheet increases alongside its uncertainty due to the strong negative feedback loop between Greenland and AMOC (Figs. N1 and N2). However, an interesting difference to the five node network including ENSO is that the Amazon rainforest can now only be influenced by the AMOC. Thus, the reduction in critical temperature is smaller than in our previous experiments, where the strong influence of ENSO impacted the Amazon rainforest's state. Now that we removed ENSO from the list of tipping elements, it is only interesting to look into specific tipping cascades of size two, but not of size three. The only remaining meaningful tipping cascade of size three is Greenland-West Antarctica-AMOC since cascades including the Amazon rainforest depend on the uncertain link between AMOC and the Amazon

**rainforest. We would like to present this robustness study in the supplementary material of the revised manuscript and discuss these findings in the paper. (Please also compare to comment to the reviewer #2)**

There are some minor points as well. 1. Line 6, "between" should be "among". 2. Line 7, insert "(AMOC)" after "Atlantic Meridional Overturning Circulation". 3. Line 35, "more simple" should be "simpler". 4. Line 86, "more simple" should be "simpler".

**Thank you very much for these minor points. These issues will be corrected in the revised version of the manuscript.**

**Reviewer #2:**

This paper presents an interesting extension of the Dekker et al. (2018) work on cascading tipping behavior by considering the possible cascading interactions of five potential tipping elements. The strength of the paper is clearly the large-scale MonteCarlo approach such that the overall behavior of the dynamical system (1) is studied. A clear weakness of the paper is the connection to climate dynamics. I suggest that the authors try to rewrite the paper to strengthen the latter aspect; the comments below are intended to help with this.

**Thank you very much for this comment. We also feel and are aware that the connection to the real climate needs better explanation: we think of our model more as some kind of hypothesis generator with which we can arrive at qualitative results (*hypotheses*) that can then, afterwards, be examined or checked by more complex models such as EMICs or GCMs. On the other side, it is and cannot be our aim with this model to make pure quantitative statements, let alone predictions.**

**Contrasting this, we see this study as a step towards more in-depth studies using more complex EMICs or GCMs. But despite the many recent progresses in EMICs and GCMs, these models would be required to resolve the nonlinear behavior of all or a sufficient subset of tipping elements which is not yet the case to our knowledge. On top of that, computational constraints might have hindered such a wide analysis as of yet which has also been mentioned for instance in Wood et al. (Clim. Dyn., 2019). These problems are overcome by our, admittedly simplified, model with its merits on the qualitative side (eg. role of tipping elements and impacts of interactions) rather than on exact quantifications. This is why we feel that the *conceptual investigation* of the Earth system with emulator like models such as this is worthwhile. We will rewrite the introduction with such a section in more detail.**

**Thus, conceptual models such as this here can infer interesting qualitative statements about the connection of tipping elements in the Earth system, such as the roles of tipping elements or the destabilisation with respect to increasing interaction strength (see Fig. 1 and 4 of the main manuscript). With that, we also contradict the possibility that the strong negative feedback between AMOC and the Greenland Ice Sheet could be a halt point for further tipping events as has been proposed earlier (Gaucherel & Moron, International J. of Climatology 2017).**

**[Compare also to comment of reviewer #1]**

1. Whereas one could justify (e.g. from conceptual models) that saddle-node bifurcations, underlying the individual dynamics term in (1), are relevant for the AMOC, ice sheets and Amazon rainforest, this does not hold for ENSO. Although this is mentioned in the paper (e.g. l63-64 and l75-78), there is no discussion on this issue. ENSO is also problematic because its behavior may not change substantially under climate change(e.g., in CMIP5 models (Kim et al.,

2014).) The best way out is probably to omit ENSO from the list of tipping elements; the results will very likely still be interesting. If the authors want to keep ENSO, they should better justify the use of (1) for this tipping element.

**We also agree that ENSO is the most controversial element within our subset of five tipping elements, we checked our results for the robustness in case ENSO is not taken into account in the list of tipping elements. The results with all simulations recomputed can be found in the new Figs. N1-N4 that were sent along with this letter and are explained in the following.**

> **The main messages regarding the role of the tipping elements still hold and, thus, are robust (the ice sheets (mainly Greenland) are initiators, the AMOC transmits cascades, see Fig. N3). Also, the critical temperature for West Antarctica and AMOC goes down, while the critical temperature for the Greenland Ice Sheet increases alongside its uncertainty due to the strong negative feedback loop between Greenland and AMOC (Figs. N1 and N2). However, an interesting difference to the five node network including ENSO is that the Amazon rainforest can now only be influenced by the AMOC. Thus, the reduction in critical temperature is smaller than in our previous experiments, where the strong influence of ENSO impacted the Amazon rainforest's state. Now that we removed ENSO from the list of tipping elements, it is only interesting to look into specific tipping cascades of size two, but not of size three. The only remaining meaningful tipping cascade of size three is Greenland-West Antarctica-AMOC since cascades including the Amazon rainforest depend on the uncertain link between AMOC and the Amazon rainforest.**
>
> **Thus, we would like to present these robustness results in the supplementary material alongside a better justification of ENSO.**

**Furthermore, we agree that ENSO as a tipping element needs better justification overall. We included the ENSO in our study since it has vital impact on other tipping elements as has already been found in Kriegler et al. (2009, PNAS), as for instance on the drying of the Amazon rainforest, especially if it is to become more frequent or even permanent (see also Duque-Villegas, 2020, ESDD).**
**In a revised version of the manuscript, we will also further elaborate on changing ENSO properties under climate change (Kim et al., 2014, Nature Climate Change, Collins et al., 2010, Nature Geoscience, Cai et al., 2014, Nature Climate Change). Whereas Kim et al. (2014) and Collins et al. (2015) emphasize the uncertainty of how ENSO will change under global warming, Cai et al. (2014) show that ENSO will increase its frequency twofold. However, certain ENSO characteristics under climate change such as an intensification of ENSO driven drying in the western Pacific and rainfall increases in the central and**

**eastern equatorial Pacific seem robust due to nonlinear responses to surface warming (Power et al., 2013, Nature).**

**Moreover, it was found that the global warming trend since the early 1990s has provided a more favorable background state for the Atlantic capacitor effect which leads to increased biennial variability in the Pacific leading to conditions that are more favorable for major El-Nino events. For this study, observational data and reanalysis data has been used (Wang et al., 2017, Nat. Communs.). More observational evidence is available from paleo data from the Pliocene (4.5 - 3.0 mio. years ago), although the Pliocene had different environmental conditions. It is hypothesized that there may have been permanent El-Nino conditions (Wara et al., 2005, Science; Ravelo et al., 2006, Gsa Today; Fedorov et al., 2006, Science). Of course the Pliocene had different environmental conditions compared to today, although the CO2 concentration is believed to be similar as today.**

**Furthermore, we feel that it is necessary to improve our explanations on why we were including ENSO in the saddle-node form if we assume that ENSO could be seen as a tipping element. The main argument is the topological equivalence (e.g. Kuznetsov et al,. 2004) of the two separated states (ENSO as it is today and a permanent ENSO) with a nonlinear reaction to changes in forcing in between. It is unclear whether ENSO would show hysteretic effects, even if we follow the argumentation above. However, since we are not investigating a possible "backtipping" to the original state, but only increasing the forcing (the global mean temperature only increases), our analysis remains valid.**

2. The coupling between the tipping elements is too sketchy at the moment and requires more discussion and analysis. The coupling terms s_{i,j} are now more or less 'guessed' but there are results of EMICs (e.g., Climber, Loveclim and modern variants) where such linear coupling coefficients could be estimated (e.g. from regression analysis). This would also shed more light on the part of the state vector (x_i in (1)) where the coupling occurs (as now only sketched in Table 2). Dekker et al. (2018) have done this to establish the relation between the AMOC and ENSO (meridional Atlantic temperature difference and the equatorial wind stress). I realize that this is more work, but it would enhance the quality and possible impact of the paper significantly.

**We agree with the reviewer that it would be great to include more sophisticated coupling terms from more complex models instead of using the s_{i,j} estimations from the expert elicitation done in Kriegler et al. (2009). In case we would aim at using such coupling terms derived from more complex models, there would arise a couple of difficulties that make their usage impossible in our view:**

**i) Despite many recent advances in GCMs and EMICs, we are unsure whether we should use models that are partially not yet able to represent the nonlinear**

behaviour of all tipping elements to further calibrate our interactions between the tipping elements with their results.

For instance the issue of ENSO representation is not yet resolved in many GCMs, tipping of the Amazon rainforest is not yet comprehensively understood and some GCMs are said to have an AMOC that is too linear. Furthermore, GCMs and EMICs mostly do not have interactive ice sheets making it difficult to estimate interaction parameters from them.

Thus, we would rather argue that we would need a new generation of models that would explicitly include the potential to investigate such nonlinear interactions.

Also, from an observational data point of view (mainly paleo data) it would be very difficult to estimate the interaction strength due to uncertainties in the relevant paleo data and the very different timescales of the functioning of the tipping elements (eg. Ice sheets on the order of millennia, Amazon rainforest much quicker on the order of tens of years).

ii) While some connections (GIS -> AMOC, AMOC -> ENSO) have been better established with EMICs like CLIMBER-2 and Loveclim as well as GCMs (Rahmstorf et al., 2005, GRL; Driesschaert et al., 2007, GRL; Sterl et al., 2008, GRL, Jungclaus et al., 2006, GRL, Wood et al., 2019, Climate Dynamics), other connections are less well established (e.g.: connections between the Greenland and West Antarctic Ice Sheet). Thus, we would feel that this would introduce a bias in the level of details of the interactions, even if we would be able to retrieve some connection terms, while others will remain unknown. However, the concerning literature can and will be cited in the revised version of the manuscript to motivate the interactions given in Kriegler et al. (2009).

iii) Interaction und individual dynamics terms would have different complexity. At the same time, it would intuitively not be clear how different physical interactions can be reduced to a comparable dimensionless interaction strength parameter for all tipping element pairs. This again would make the experiments difficult where we scale up the interaction parameter d to investigate tipping temperatures (Figs 4 and 5) and the role of tipping elements (Fig. 1).

Following this, we agree that our study should be seen as the basis of more in-depth investigations of tipping interactions. Hence, we feel that this would be beyond the scope for this paper, although some few interaction links have been investigated with conceptual models, EMICs or GCMs (e.g., Dekker et al., 2018, ESD, Wood et al., 2019, Clim. Dyn., Rahmstorf et al., 2005, GRL, Hawkins et al., 2011, GRL).

**A last remark to the expert elicitation from Kriegler et al. (2009):**

> **In such an expert elicitation, there are of course uncertainties and they are reflected in the large spread of the estimated values. On the other hand, at this expert elicitation, there have been leading experts for each of the five tipping elements. Thus, we assume that the expert elicitation is more than "guess-work", although there are large uncertainties. For instance, the increase of the likelihood of tipping of the AMOC in response to a tipping Greenland Ice Sheet is increased by a factor of 1 to 10. This is a large spread. This interval and the intervals from the other links between the tipping elements are taken into account into the large scale Monte Carlo simulation propagating these uncertainties.**

3. With ENSO removed and a better justification of the linear coupling from EMIC results (points 1 and 2 above), the interpretation of the results in Fig. 4-6 can be much improved. This in particular holds for the interesting result that the coupling destabilizes the reference climate state (as mentioned in the paper l268-270). Section 4 can then be substantially improved and it would be particularly helpful to the community if suggestions would be given on climate model experiments (even with EMICs) which could test the occurrence of this cascading behavior. There are several more minor issues but as the paper probably is rewritten substantially with different results, I will not mention these here.

**We agree that our conclusion (Section 4) can be much improved with the new set of simulations and robustness checks of all our results without ENSO. We will rewrite this section in a revised version of our manuscript.**

**Potential investigations with EMICs, GCMs or conceptual models could be very helpful. Some possible examples are:**

1. **One such experiment could be the investigation of changing precipitation patterns over Amazonia due to a tipped AMOC, i.e., whether rainfall will increase or decrease and whether this would potentially be sufficient to induce a tipping cascade. This would shed light on the interaction pair AMOC-Amazon rainforest. This could potentially also be extended to the tipping chain of a melting Greenland Ice Sheet, its influence on AMOC which then impacts the Amazon rainforest. This could be done by hosing experiments as described in Wood et al. (2019, Clim. Dyn.)**

2. **Also, the influence of the disintegration of the West Antarctic Ice Sheet on the AMOC could be investigated by introducing freshwater input into the AMOC around the West Antarctic Ice Sheet. Then one could observe the reaction of AMOC under different hosing parameters (amount of freshwater input) as has already often been done for Greenland and AMOC. There exist already some**

comparison experiments of run-off from Greenland and West Antarctica (Ivanovic et al., GRL, 2018), but we have not found isolated experiments on this.

3. Then, further: if the EMIC would have interactive ice sheets, it would be possible to investigate the tipping triplet GIS-AMOC-WAIS to investigate the impact of the Greenland Ice Sheet on West Antarctica and vice versa. This could for instance be done by hosing experiments around Greenland or West Antarctica (or both).

Furthermore, it might be worthwhile to perform a new expert elicitation on the connection pattern (feedbacks) between the tipping elements, but also on the set of tipping elements itself. We will include such ideas and suggestions for ESM model experiments into our new version of the manuscript.

**Results without ENSO as a tipping element**

[Figure]

**Fig. N1 (compare to Fig. 4 of the main manuscript): Shift of critical temperature ranges due to interactions omitting ENSO. (a) Critical global mean temperatures for each of the four investigated tipping elements, without taking interactions into account (as reproduced from literature (Schellnhuber et al. (2016), Nat. Clim. Change). The grey bars indicate the standard deviation arising when drawing from a random uniform distribution between the respective upper and lower temperature limits. These bars correspond to the critical temperature ranges in case of zero interaction strength in panels b and c. (b, c) Change of critical temperature ranges with increasing interaction strength for the Greenland Ice Sheet and West Antarctic Ice Sheet (panel b) and the Atlantic Meridional Overturning Circulation (AMOC) and Amazon rainforest (panel c). The standard deviation of the critical temperatures for each tipping element within the Monte Carlo ensemble is given as respective colour shading.**

[Figure]

**Fig. N2 (compare to Fig. 5 of the main manuscript): Difference in critical temperatures with respect to the interaction strength without ENSO as a tipping element. Difference of critical temperatures in °C (left panels) and % (right panels) compared to the respective initially drawn critical temperature for the four investigated tipping elements: (a, b) Greenland Ice Sheet, (c, d) West Antarctic Ice Sheet, (e, f) AMOC, (g, h) Amazon rainforest. The standard deviation from the ensemble members is shown as respective colour shading**

[Figure]

**Fig. N3 (compare to Fig.1 and Fig.S1 of the main manuscript and supplement):** *Role* of tipping elements in cascades without ENSO as a tipping element. a) Relative frequency in percent of occurrence of a certain tipping element in a tipping cascade (hatched bars). The standard deviation is computed by evaluating the deviation between reasonable network settings. b) Relative frequency in percent that a certain tipping element causes a tipping cascade (coloured bars). We define that the cause of a cascade is the element, whose critical temperature is closest to the temperature of the cascade. Again the error bars show the standard deviation between different network settings as in a. c) Count versus global mean temperature increase at which a tipping cascade occurs divided into the respective five tipping elements. d) Same as in c, but for the tipping element which causes the cascade. N.B.: Panel c) and d) are set to the same scale normalised to the highest value in the histogram.

Note also that the Amazon rainforest cannot be an initiator of cascades anymore since it does not influence any other tipping element, but is only influenced (by the AMOC). [Initiators: 65+-2% (GIS), 23+-3% (WAIS), 12+-2% (AMOC), 0% (AMAZ)

Occurrence: 29+-1% (GIS), 31+-2% (WAIS), 28+-2% (AMOC), 11+-2%(AMAZ)]

[Figure]

**Fig. N4 (compare with Fig. 6 of the main manuscript): Tipping cascades without ENSO as a tipping element. (a, b)** For global warming up to 2.0°C above pre-industrial, the colour shading illustrates the fraction of model representations in the Monte-Carlo ensemble without tipping events (grey), with a singular tipping event (purple) and with cascades including two (red), three (dark orange) and four (orange) elements. **(c, d, e)** Occurrence of tipping cascades of size two, three and four versus global mean temperature increase. The counts are normalised to the highest value of the most frequent tipping cascade (in cascades of size two). Tipping cascades of size three and four are set to the same scale to secure comparability. **(f),** Dominant cascades of size two/three for temperature increases from 0-8°C above pre-industrial. Other cascades are not shown, since their relative occurrence is much smaller than for the ones shown. The standard deviation represents the difference between the nine different network settings. Uncertainties are larger for network representations, where unclear links are involved, e.g., for the AMOC-Amazon rainforest tipping pair (compare Fig. 1 of the main manuscript).

---

## Editor Comment (EC1) · Michel Crucifix (Editor) · 25 Jun 2020

Dear authors,

I would like to thank both reviewers and authors for the overall constructive discussion phase. I am particularly pleased to read that the authors will additional sensitivity studies to the role of the ENSO bifurcation, and also clarify your justification for adopting a fold-bifurcation model, which indeed was probably not entirely clear in the original version of the manuscript. I therefore invite the authors to provide their revised version.
* * *

---

## Author Response (AR1)

Dear Mr. Crucifix, Dear Reviewers,

We thank the reviewers for their reports on our manuscript and the editor for the opportunity to improve our manuscript. The comments help us to sharpen and improve our research paper substantially.

The major revisions and changes in the manuscript in response to the comments from the reviewers comprise:

1. Structural robustness analysis with respect to ENSO for all results and in-depth discussion about ENSO as a tipping element
2. Highlight of the usefulness of the conceptual approach in the introduction and methods: we view our approach as a hypotheses generator and a basis for more process-detailed studies
3. Careful integration of further literature sources, especially with regard to the motivation of the interaction structure for known interactions
4. Discussion about experiments with more complex models on interaction paris that could be investigated (in the conclusion and discussions)

Please find below a detailed point-by-point response to the reviewer comments. We also attached the new version of our manuscript and supplement below and marked the changes in blue.

We are thankful for the opportunity to improve our manuscript and are looking forward to further feedback.

Sincerely yours,
Nico Wunderling, Jonathan Donges, Jürgen Kurths & Ricarda Winkelmann

**Reviewer #1:**

The authors are discussing domino effects of individual tipping elements when the elements are coupled. The authors are investigating a set of coupled systems for evaluating how tipping phenomena cascade in the coupled systems. My major concern is that although the authors succeeded in showing how tipping phenomena cascade in the toy model, the set of coupled systems cannot imply anything related to the actual climate, in my opinion, because the set of coupled systems is too conceptual to infer something in a real climate. Thus, the authors should discuss this gap more carefully to clarify the limitations as well as surely ensured implications of the current study. This is my major concern.

**We agree with the reviewer that our model is based on a couple of simplifications that should be made clearer with respect to its limitations in the manuscript. Opposed to a toy model, we view our model as a hypotheses generator with which we can arrive at interesting and valuable qualitative results (*hypotheses*) that can then, afterwards, be examined or checked by more complex models such as EMICs or GCMs. On the other side, it is and cannot be our aim with this model to make pure quantitative statements, let alone predictions.**

**Contrasting this, we see this study as a step towards more in-depth studies using more complex EMICs or GCMs. But despite the many recent progresses in EMICs and GCMs, these models would be required to resolve the nonlinear behavior of all or a sufficient subset of tipping elements which is not yet the case to our knowledge. On top of that, computational constraints might have hindered such a wide analysis as of yet which has also been mentioned for instance in Wood et al. (Clim. Dyn., 2019). These problems are overcome by our, admittedly simplified, model with its merits on the qualitative side (eg. role of tipping elements and impacts of interactions) rather than on exact quantifications or even predictions. This is why we feel that the *conceptual investigation* of the Earth system with emulator-like models such as this is worthwhile and creates interesting insights into the dynamics of interacting tipping elements. Also because there has been stated that some interactions might stabilise the Earth system (Gaucherel, Int. J. Climatol., 2017), while other studies hypothesise a considerable risk in tipping cascades up to a potential global cascade (Steffen et al., PNAS, 2018; Lenton et al., Nature, 2019). We have reformulated and extended major parts of the introduction (see ll 31-60).**

**In our manuscript, the main qualitative findings are:**

1. **We find that the ice sheets are the tipping elements which are most likely to initiate cascades of tipping events. Especially tipping cascades from the Greenland Ice Sheet to the AMOC as well as (less, but also significant) from the West Antarctic Ice Sheet to the AMOC fall into this category (see Fig. 1 and Fig. 6e,f of the main manuscript). This is also supported by many literature studies with conceptual models (eg. Wood et al., Clim. Dyn. 2019, Stommel, Tellus 1961,**

etc.), where the AMOC is influenced by freshwater input, mainly from a melting Greenland Ice Sheet. Furthermore, a reduction of the AMOC has also been found in data and general circulation models (Caesar et al., Nature 2018, Rahmstorf et al., Nat. Clim. Change, 2015, Hawkins et al., GRL 2011). We consider this result as very robust and physically meaningful.

2. While the ice sheets are the initiators of tipping cascades, we found that the AMOC is a mediator/transmitter of cascades. In our model, this results merely comes from the fact that the AMOC has the most connections to other tipping elements (i.e., from a topological point of view). But also in the real climate system, the AMOC connects the two polar regions via the equator due to its specific structure influenced by the melting ice from Greenland as well as from Antarctica (eg. Swingedouw et al., Clim. Dyn. 2009, Hu et al., Journal of Climate 2013). Thus, we think that this role as a connector would remain in a more process based study with, for instance, GCMs (see section 3.3 and ll 325-330).

3. Furthermore, we think that the reduction of the critical temperatures for all tipping elements but Greenland is also a robust result that would hold under further studies (see Fig. 4 and Fig. 5).

In a revised version of the manuscript, we would emphasize these points and write it in a clearer way, also emphasizing that future research should aim at attaining more details on the specific interaction pairs (see ll 360-385).

On top of that, we performed a structural robustness analysis and recomputed all our results without ENSO since it is debated whether ENSO is a tipping element and how a tipped state might play out with climate change, e.g.: will ENSO be permanent, will it be stronger, …? (see ll 83-105 and ll 121-140)
Thus, to improve our model, we recomputed all our results to check the robustness without taking ENSO into account as a tipping element (see supplementary Figs. S3 to S6).

The main messages regarding the role of the tipping elements still hold and, thus, are robust (the ice sheets (mainly Greenland) are initiators, the AMOC transmits cascades, see Fig. S3). Also, the critical temperature for West Antarctica and AMOC goes down, while the critical temperature for the Greenland Ice Sheet increases alongside its uncertainty due to the strong negative feedback loop between Greenland and AMOC (Figs. S4 and S5). However, an interesting difference to the five node network including ENSO is that the Amazon rainforest can now only be influenced by the AMOC. Thus, the reduction in critical temperature is smaller than in our previous experiments, where the strong influence of ENSO impacted the Amazon rainforest's state.

**We present the figures of this robustness study in the supplementary material of the revised manuscript and discuss their findings in the paper (see ll 331-343 and supplementary Figs. S3 to S6; please also compare to comment to the reviewer #2).**

There are some minor points as well. 1. Line 6, "between" should be "among". 2. Line 7, insert "(AMOC)" after "Atlantic Meridional Overturning Circulation". 3. Line 35, "more simple" should be "simpler". 4. Line 86, "more simple" should be "simpler".

**Thank you very much for these minor points. These issues have been corrected in the revised version of the manuscript.**

**Reviewer #2:**

This paper presents an interesting extension of the Dekker et al. (2018) work on cascading tipping behavior by considering the possible cascading interactions of five potential tipping elements. The strength of the paper is clearly the large-scale MonteCarlo approach such that the overall behavior of the dynamical system (1) is studied. A clear weakness of the paper is the connection to climate dynamics. I suggest that the authors try to rewrite the paper to strengthen the latter aspect; the comments below are intended to help with this.

**Thank you very much for this comment. We also feel and are aware that the connection to the real climate needs better explanation: we think of our model more as a hypotheses generator with which we can arrive at qualitative results (*hypotheses*) that can then, afterwards, be examined or checked by more complex models such as EMICs or GCMs. This seems also important to us because it has been stated that some interactions might stabilise the Earth system (Gaucherel, Int. J. Climatol., 2017), while other studies hypothesise a considerable risk in tipping cascades up to a potential global cascade (Steffen et al., PNAS, 2018; Lenton et al., Nature, 2019). With our model, we are able to check such hypotheses. On the other side, we do not aim at making predictions.**

**We see this study as a step towards more in-depth studies using more complex EMICs or GCMs. But despite the many recent progresses in EMICs and GCMs, these models would be required to resolve the nonlinear behavior of all or a sufficient subset of tipping elements which is not yet the case to our knowledge. On top of that, computational constraints might have hindered such a wide analysis as of yet which has also been mentioned for instance in Wood et al. (Clim. Dyn., 2019). These problems are overcome by our, admittedly simplified, model with its merits on the qualitative side (eg. role of tipping elements and impacts of interactions) rather than on exact quantifications. This is why we feel that the *conceptual investigation* of the Earth system with emulator like models such as this is worthwhile. We have rewritten the respective parts in the introduction (see ll. 31-60).**

**Thus, conceptual models such as this here can infer interesting qualitative statements about the connection of tipping elements in the Earth system, such as the roles of tipping elements or the destabilisation with respect to increasing interaction strength (see Fig. 1 and 4 of the main manuscript). With that, we also contradict the possibility that the strong negative feedback between AMOC and the Greenland Ice Sheet could be a halt point for further tipping events as has been proposed earlier (Gaucherel & Moron, International J. Climatol., 2017, see ll 358-359).**

**[Compare also to comment of reviewer #1]**

1. Whereas one could justify (e.g. from conceptual models) that saddle-node bifurcations, underlying the individual dynamics term in (1), are relevant for the AMOC, ice sheets and

Amazon rainforest, this does not hold for ENSO. Although this is mentioned in the paper (e.g. l63-64 and l75-78), there is no discussion on this issue. ENSO is also problematic because its behavior may not change substantially under climate change(e.g., in CMIP5 models (Kim et al., 2014).) The best way out is probably to omit ENSO from the list of tipping elements; the results will very likely still be interesting. If the authors want to keep ENSO, they should better justify the use of (1) for this tipping element.

**We also agree that ENSO is the most controversial element within our subset of five tipping elements. We checked our results for robustness in case ENSO is not taken into account in the list of tipping elements for all results. The results with all simulations can be found in the new supplementary Figs. S3-S6.**

> **The main messages regarding the role of the tipping elements still hold and, thus, are robust (the ice sheets (mainly Greenland) are initiators, the AMOC transmits cascades, see Fig. S3). Also, the critical temperature for West Antarctica and AMOC goes down, while the critical temperature for the Greenland Ice Sheet increases alongside its uncertainty due to the strong negative feedback loop between Greenland and AMOC (Figs. S4 and S5). However, an interesting difference to the five node network including ENSO is that the Amazon rainforest can now only be influenced by the AMOC. Thus, the reduction in critical temperature is smaller than in our previous experiments, where the strong influence of ENSO impacted the Amazon rainforest's state.**
> **We present these robustness results in the supplementary material and discuss the implications in the main manuscript (see ll 331-343 and supp. Figs. S3-S6).**

**Alongside, we provide a better justification of ENSO since we agree that ENSO as a tipping element needs better justification overall. This discussion can be found below and in the manuscript in ll 83-105.**

**We included the ENSO in our study since it has vital impact on other tipping elements as has already been found in Kriegler et al. (2009, PNAS), as for instance on the drying of the Amazon rainforest, especially if it is to become more frequent or even permanent (see also Duque-Villegas, 2020, ESDD).**
**In the new version of the manuscript, we discuss in much more detail how ENSO might or might not change under global warming scenarios (Kim et al., 2014, Nature Climate Change, Collins et al., 2010, Nature Geoscience, Cai et al., 2014, Nature Climate Change). Whereas Kim et al. (2014) and Collins et al. (2010) emphasize the uncertainty of how ENSO will change under global warming, Cai et al. (2014) find that ENSO will increase its frequency twofold. However, certain ENSO characteristics under climate change such as an intensification of ENSO driven drying in the western Pacific and rainfall increases in the central and eastern equatorial Pacific seem robust due to nonlinear responses to surface warming (Power et al., 2013, Nature).**

**Moreover, it was found that the global warming trend since the early 1990s has provided a more favorable background state for the Atlantic capacitor effect which leads to increased biennial variability in the Pacific leading to conditions that are more favorable for major El-Nino events. For this study, observational data and reanalysis data has been used (Wang et al., 2017, Nat. Communs.). More observational evidence is available from paleo data from the Pliocene (4.5-3.0 mio. years ago). It is hypothesized that there may have been permanent El-Nino conditions (Wara et al., 2005, Science; Ravelo et al., 2006, Gsa Today; Fedorov et al., 2006, Science). Of course the Pliocene had different environmental conditions compared to today, although the CO2 concentration is believed to be similar to today.**

**Furthermore, we feel that it is necessary to improve our explanations on why we were including ENSO in the saddle-node form when assuming that ENSO can be seen as a tipping element. The main argument is the topological equivalence (e.g. Kuznetsov et al,. 2004) of the two separated states (ENSO as it is today and a permanent ENSO) with a nonlinear reaction to changes in forcing in between. It is unclear whether ENSO would show hysteretic effects, even if we follow the argumentation above. However, since we are not investigating a possible "backtipping" to the original state, but only increasing the forcing (the global mean temperature only increases), we can make this simplification (see ll 121-140).**

2. The coupling between the tipping elements is too sketchy at the moment and requires more discussion and analysis. The coupling terms s_{i,j} are now more or less 'guessed' but there are results of EMICs (e.g., Climber, Loveclim and modern variants) where such linear coupling coefficients could be estimated (e.g. from regression analysis). This would also shed more light on the part of the state vector (x_i in (1)) where the coupling occurs (as now only sketched in Table 2). Dekker et al. (2018) have done this to establish the relation between the AMOC and ENSO (meridional Atlantic temperature difference and the equatorial wind stress). I realize that this is more work, but it would enhance the quality and possible impact of the paper significantly.

**We agree with the reviewer that it would be great to include more sophisticated coupling terms from more complex models instead of using the s_{i,j} estimations from the expert elicitation done in Kriegler et al. (2009). In case we would aim at using such coupling terms derived from more complex models, there would arise a couple of difficulties that make their usage impossible in our view (We also succinctly discuss this in the revised version of the manuscript in ll 182-194):**

**i) Despite many recent advances in GCMs and EMICs, we are unsure whether we should use models that are partially not yet able to represent the nonlinear**

behaviour of all tipping elements to further calibrate our interactions between the tipping elements with their results.

For instance the issue of ENSO representation is not yet resolved in many GCMs, tipping of the Amazon rainforest is not yet comprehensively understood and some GCMs are said to have an AMOC that is too linear. Furthermore, GCMs and EMICs mostly do not have interactive ice sheets making it difficult to estimate interaction parameters from them.

Thus, we would rather argue that we would need a new generation of models that would explicitly include the potential to investigate such nonlinear interactions.

Also, from an observational data point of view (mainly paleo data) it would be very difficult to estimate the interaction strength due to uncertainties in the relevant paleo data and the very different timescales of the functioning of the tipping elements (eg. Ice sheets on the order of millennia, Amazon rainforest much quicker on the order of tens of years).

ii) While some connections (GIS -> AMOC, AMOC -> ENSO) have been better established with EMICs like CLIMBER-2 and Loveclim as well as GCMs (Rahmstorf et al., 2005, GRL; Driesschaert et al., 2007, GRL; Sterl et al., 2008, GRL, Jungclaus et al., 2006, GRL, Wood et al., 2019, Climate Dynamics), other connections are less well established (e.g.: connections between the Greenland and West Antarctic Ice Sheet). Thus, we would feel that this would introduce a bias in the level of details of the interactions, even if we would be able to retrieve some connection terms, while others will remain unknown. However, the concerning literature can and will be cited in the revised version of the manuscript to motivate the interactions given in Kriegler et al. (2009).

iii) Interaction und individual dynamics terms would have different complexity. At the same time, it would intuitively not be clear how different physical interactions can be reduced to a comparable dimensionless interaction strength parameter for all tipping element pairs. This again would make the experiments difficult where we scale up the interaction parameter d to investigate tipping temperatures (Figs. 4 and 5) and the role of tipping elements (Fig. 1).

Following this, we agree that our study should be seen as the basis of more in-depth investigations of tipping interactions. Hence, we feel that this would be beyond the scope for this paper, although some few interaction links have been investigated with conceptual models, EMICs or GCMs (e.g., Dekker et al., 2018, ESD, Wood et al., 2019, Clim. Dyn., Rahmstorf et al., 2005, GRL, Hawkins et al., 2011, GRL, Hu et al., 2013, J. Climate).

**A last remark to the expert elicitation from Kriegler et al. (2009):**

> In such an expert elicitation, there are of course uncertainties and they are reflected in the large spread of the estimated values. On the other hand, at this expert elicitation, there have been leading experts for each of the five tipping elements. Thus, we think that the expert elicitation is more than "guess-work", although there are large uncertainties. For instance, the increase of the likelihood of tipping of the AMOC in response to a tipping Greenland Ice Sheet is increased by a factor of 1 to 10. This is a large spread. This interval and the intervals from the other links between the tipping elements are taken into account into the large scale Monte Carlo simulation propagating such uncertainties.

3. With ENSO removed and a better justification of the linear coupling from EMIC results (points 1 and 2 above), the interpretation of the results in Fig. 4-6 can be much improved. This in particular holds for the interesting result that the coupling destabilizes the reference climate state (as mentioned in the paper l268-270). Section 4 can then be substantially improved and it would be particularly helpful to the community if suggestions would be given on climate model experiments (even with EMICs) which could test the occurrence of this cascading behavior. There are several more minor issues but as the paper probably is rewritten substantially with different results, I will not mention these here.

**We agree that our conclusion (Section 4) can be much improved with the new set of simulations and robustness checks of all our results without ENSO and potential investigations with EMICs, GCMs or conceptual models could be very helpful. Some possible examples could comprise:**

1. **One such experiment could be the investigation of changing precipitation patterns over Amazonia due to a tipped AMOC, i.e., whether rainfall will increase or decrease and whether this would potentially be sufficient to induce a tipping cascade. This would shed light on the interaction pair AMOC-Amazon rainforest. This could potentially also be extended to the tipping chain of a melting Greenland Ice Sheet, its influence on AMOC which then impacts the Amazon rainforest. This could for instance be done by hosing experiments as described in Wood et al. (2019, Clim. Dyn.).**

2. **Also, the influence of the disintegration of the West Antarctic Ice Sheet on the AMOC could be investigated by introducing freshwater input into the AMOC around the West Antarctic Ice Sheet. Then one could observe the reaction of AMOC under different hosing parameters (amount of freshwater input) as has already often been done for Greenland and AMOC. There exist already some comparison experiments of run-off from Greenland and West Antarctica (Ivanovic**

et al., GRL, 2018; Hu et al., J. Climate, 2013), but a comprehensive study, maybe including dynamic ice sheets will shed more light on this interaction pair.

3. Then, further: if the EMIC would have interactive ice sheets, it would be possible to investigate the tipping triplet GIS-AMOC-WAIS to investigate the impact of the Greenland Ice Sheet on West Antarctica and vice versa.

Furthermore, it might be worthwhile to perform a new expert elicitation on the connection pattern (feedbacks) between the tipping elements, but also on the set of tipping elements itself. We included these ideas in the manuscript in ll 368-385.

[revised manuscript text omitted]

**Structural robustness analysis without ENSO**

In this section, we perform a structural robustness analysis without taking ENSO into account as a tipping element since it is debated whether and to which extent ENSO should be seen as a tipping element (discussion see main manuscript).

[Figure]

**Figure S 3.** *Role* of the tipping elements without ENSO. The panels show the same as in Fig. S1 and the dominos in Fig. 1 (there with ENSO). It is found that the Greenland Ice Sheet is a dominant initiator of tipping cascades (65±2%), while West Antarctice (23±3%) and AMOC (12±2%) initiate less cascades. At the same time, the occurrence of the elements within cascades is relatively similar for the Greenland Ice Sheet (29±1%), the West Antarctic Ice Sheet (31±2%) and the AMOC (28±2%). Thus, the West Antarctic Ice Sheet can also be seen as an intermediate initiator of cascades, while the AMOC remains a dominant mediator of cascades. Note that the Amazon rainforest has a special role now since it is only connected to the AMOC (via an unclear link). Thus, in many cascades the Amazon rainforest does not occur (overall occurrence: 11±2%) and it cannot be the reason for a tipping cascade since the number of outgoing links is zero. Thus, the Amazon rainforest is not depicted in panels (**b**) and (**d**).

[Figure]

**Figure S 4.** Shift of critical temperature ranges due to interactions without ENSO. The panels are the same as in Fig. 4 of the main manuscript including ENSO. With increasing interaction strength, the critical temperatures develop similar as in the case with ENSO for the Greenland Ice Sheet, the West Antarctic Ice Sheet and the AMOC. For the Amazon rainforest, the reduction of its critical temperature is small since the only connection consists in the uncertain link from AMOC to the Amazon rainforest.

[Figure]

**Figure S 5.** Difference in critical temperatures with respect to the interaction strength without ENSO. The panels show the same as in Fig. 5 of the main manuscript with ENSO. The results are similar apart from the Amazon rainforest that shows less reduction in its critical temperature due to its lose connection to the other three tipping elements via one unclear link from the AMOC.

[Figure]

**Figure S 6.** Tipping cascades without ENSO. The results with ENSO are shown in the Figs. 6 and S2. The warming levels of tipping cascades are very similar to the simulations that include ENSO, including the fact that cascades could occur for global warming levels below 2° above pre-industrial similar to the results in the main manuscript (see Fig. 6a, b).

**References**

5    Baudin, M.: pyDOE: The experimental design package for python, sofware available under the BSD license (3-Clause) at https://pythonhosted.org/pyDOE/index.html, 2013.

---

## Author Response (AR2)

Dear Mr. Crucifix, Dear Reviewers,

We are grateful for the comments by the reviewer and the editor. We appreciate the very helpful suggestions which helped to further clarify and improve our manuscript.

We have now addressed all comments. Major changes in the revised manuscript include:

1. Following the reviewer's suggestions, we have excluded ENSO from our analysis in the main manuscript and now discuss it as an additional structural robustness analysis in the supplementary information.
2. We have clarified and substantiated our arguments why each of the remaining four tipping elements can be modelled with the cusp-equation as in our approach and added further literature sources.
3. Further, we more broadly elaborated on the literature concerning the interactions between the four tipping elements, both from modelling and observation studies.
4. We have restructured the introduction and methods part to improve the presentation in line with the reviewers and editors comments.

Please find below a detailed point-by-point response to the comments. We also attached the new version of our manuscript and supplement below and marked the changes in blue.

We are grateful for the opportunity to improve our manuscript and are looking forward to further feedback.

Sincerely yours,
Nico Wunderling, Jonathan Donges, Jürgen Kurths & Ricarda Winkelmann

**Editor's comments**

We are grateful to the editor for this highly valuable summary. We have revised the manuscript thoroughly and extensively accordingly. Please find below a short summary of our response to the editor's comments. A more detailed answer is posted below as a reply to the comments from the reviewer. Since major parts of the revisions require additional literature sources, a reference list can be found at the end of this response letter.

Dear authors,

I have only received one review of the revised version of your manuscript, but I believe this will suffice to proceed with a decision. As you will see the reviewer still expresses some concerns, and has asked to see the paper again. I believe, though, that it should be possible to proceed reasonably swiftly, and let me comment on the concerns of this reviewer.

- The representation of the tipping point of ENSO is "inadequate". As a matter of fact, we had a reading club in my group around your paper and we came up with a similar concern. I would therefore support the reviewer's suggestion to emphasise the ENSO-free case, and consider the additional ENSO tipping element as a sensitivity experiment.

We agree with the editor that the representation of ENSO is not adequately represented by Equation (1). After careful consideration, we have therefore followed your advice and excluded ENSO from the main part of the manuscript and moved the corresponding analysis to the supplementary information as a thorough structural robustness analysis. Instead, we now show the case without ENSO as a tipping element in the main manuscript. Please see below for a more detailed answer.

- Regarding the ice sheet dynamics: I am less sure to follow the reviewer here, since actually you referred to the Leverman - Winkelman article which, with its Figure 1, seems to answer the reviewer comment. This said, the contributions by Ch. Schoof are worth citing. More generally, I understand the reviewer's concern that citing early assessments such as Kriegler's or Lenton's might take more from these papers than what they actually meant to provide: their objective was to provide a perspective based on available and sometimes fragmentary evidence. They would not stand as a justification for the bifurcation structure to be associated with the tipping elements being considered here.

We are very thankful for these considerations and we think that the manuscript has now substantially improved by the inclusion of additional evidence from literature. These sources justify the type of bifurcation used in this manuscript and give extra details on the interactions between the tipping elements. For both parts, we included evidence from conceptual models, but also performed an extensive literature research, also building on

**literature from existing more complex models that found such a bifurcation type or clues for interactions between the tipping elements. For more details, please see our answer to the reviewer comments.**

- Please consider all minor comments.

**We answered all minor points.**

Most sincerely,
Michel Crucifix

**Reviewer 1 comments**

In response to my earlier review, the authors have added the cascading analysis without ENSO (in the SM) and have added more argumentation on why ENSO is considered, and to support the coupling of the different tipping elements. The paper has improved, but I think it still needs a round of revision to make it suitable for publication in Earth System Dynamics, following the comments below.

Major:

1. The justification of modelling ENSO through an equation (1) is not adequate. In the deterministic case, the ENSO transition is between a fixed point and a limit cyle. In the limit cycle, El Nino's occur but also La Nina's and although the mean state is slightly skewed towards the warm phase, the effect of the oscillation on the mean state (and hence on the coupling to the other tipping elements) is rather small. The permanent El Nino state in the Pliocene is much debated as many GCMs just show El Nino variability during this period, and it is not relevant for the present climate change context as the geometry of the Pacific was different (open Panama gateway). As the results for the case without ENSO (Figs. S3-S5) are not essentially diffferent (apart from the Amazon Forest behavior in Fig. S5), the paper would be much better when the results without ENSO would appear in the main text and the results with ENSO are only discussed in the last section of the paper, mentioning the caveat that one cannot justify (1) for ENSO.

**We thank the reviewer for this insightful comment and additional explanation. After careful consideration, we have followed the advice and excluded ENSO from the main manuscript and moved the corresponding analysis to the supplementary information as a structural robustness analysis. As advised, we only discuss the results including ENSO in the last section of our paper with a statement that Eq. (1) is not entirely appropriate for describing ENSO for the reasons put forward by the reviewer. In the revised manuscript, all figures have been adapted accordingly and large parts of the discussion on ENSO have been shifted to the supplement and the last part of the results section. The structural robustness analysis including ENSO are described in Section 3.4 (ll 425-470) and the supplementary material. In all other parts in the main manuscript, only results without ENSO are shown. Due to the exclusion of the ENSO node in the network, the ensemble size that we compute is reduced from 11 mio. ensemble members to 3.7 mio. ensemble members since the link from the Amazon rainforest to ENSO is not part of our uncertainty propagation anymore.**

2. In section 2, it should at least be argued very well (i) why each of the tipping elements considered can be represented by an equation (1), e.g. based on conceptual models, and (ii) that there are indications from models and/or observations of coupling between the tipping elements with a plausible physical description.

Issue (i) is mentioned in lines 116-124, but the references for the MIS are rather sparse (e.g. papers Weertman, Schoof) so this needs to be extended.

**We truly appreciate this suggestion and believe that including the additional evidence for each of the tipping elements and its respective representation via equation (1) has massively improved the manuscript (see Section 2.1 in ll 127-194). In the following, we outline our line of argumentation:**

**First, we are well-aware that the representation of a complex climate tipping element with all its interacting processes as well as positive and negative feedbacks in a single cusp bifurcation is an immense simplification.**

**Nonetheless, we would argue that for our purpose of studying the role of the interaction network and strengths, the overall structure of the remaining four elements (AMOC, Greenland Ice Sheet, West Antarctic Ice Sheet, Amazon rainforest) can be well-represented by a cusp bifurcation.**

**Based on the extensive body of literature, we assume each of these to be a tipping element of the climate system. Conceptual models and basic physical understanding of the feedbacks explicitly show this cusp structure for climate tipping elements (Bathiany et al., 2016, Dynam. Stat. Clim. Syst.), and also many more complex, process-based and highly resolved models indicate this.**

**In the following, we describe the four tipping elements considered here. The same argumentation can be found in the manuscript in Section 2.1.**

**AMOC:**
**Early conceptual models introduced in the 1960ies showed that the AMOC can exhibit a cusp-like behaviour, using simplified box models based on the so-called salt-advection feedback (Stommel, 1961, Tellus; Cessi, 1994, J. Phys. Oceanogr.). Many extensions and updates to the well-known box model approach have been put forward, each confirming the potential multi-stability of the AMOC (e.g. Wood et al., 2019, Clim. Dynam.). More complex Earth system models including EMICs (e.g., CLIMBER) and AOGCMs (e.g., the FAMOUS and HadGEM3 models) have shown hysteresis behaviour which is qualitatively similar to Eq. 1 (Rahmstorf et al., 2005, Geophys. Res. Lett.; Hawkins et al., 2011, Geophys. Res. Lett.; Mecking et al., 2016, Clim. Dynam.). Furthermore, paleoclimatic evidence suggests a bistability of the AMOC: In paleoclimate records, Dansgaard-Oeschger events (see e.g. Crucifix, 2012, Philos. Trans. R. Soc. A) have been**

associated with large reorganisations of the AMOC (Ganopolski and Rahmstorf, 2002, Phys. Rev. Lett.; Timmermann et al., 2003, J. Climate; Ditlevsen et al., 2005, J. Climate), where ice core data links the events to sea-surface temperature increases in the North Atlantic. Even though there are considerable uncertainties, literature estimates the level of global warming sufficient for tipping the AMOC between 3.5–6.0 C (Schellnhuber et al., 2016, Nat. Clim. Change; Lenton, 2012, Ambio; Levermann et al., 2012, Clim. Change; Lenton et al., 2008, Proc. Natl. Acad. Sci.), considerably increasing above 4°C above pre-industrial levels (Kriegler et al., 2009, Proc. Natl. Acad. Sci.).

From these reasons, we think that Eq. (1) can be justified for the AMOC.

**GREENLAND & WEST ANTARCTIC ICE SHEETS:**
Previous studies have shown that multistability and the cusp-like structure can result from the melt-elevation feedback (Levermann and Winkelmann, 2016, The Cryosphere) as well as from the MISI (Schoof, 2007, J. Geophys. Res.-Earth; Weertman, 1974, J. Glaciol.; DeConto & Pollard, 2016, Nature).

**Greenland Ice Sheet:**
Previous studies have shown that a fold-bifurcation structure for the ice sheets can arise from the melt-elevation feedback (Levermann & Winkelmann, 2016, The Cryosphere) as well as from the Marine Ice Sheet Instability and other positive feedback mechanisms (e.g., DeConto & Pollard, 2016, Nature; Schoof, 2007, J. Geophys. Res.-Earth; Weertman, 1974, J. Glaciol.). In particular, dynamic ice sheet model simulations have identified irreversible ice loss once a critical temperature threshold is crossed (Toniazzo et al., 2004, J. Climate), leading to multiple stable states and hysteresis behaviour for the Greenland Ice Sheet (Robinson et al., 2012, Nat. Clim. Change; Ridley et al., 2010, Clim. Dyn.). In Robinson et al. (2012), the critical temperature range for an irreversible disintegration of the Greenland Ice Sheet has been estimated between 0.8–3.2 C of warming above pre-industrial levels. Paleoclimate evidence further suggests that there have been substantial, potentially self-sustained retreats of the Greenland Ice Sheet in the past. It has, for instance, been simulated that the Greenland Ice Sheet can become ice-free in case presumably warmer ocean conditions from the Pliocene are applied to an initially glaciated Greenland (Koenig et al., 2014, Geophys. Res. Lett.). Further, Greenland was nearly ice-free for extended interglacial periods during the Pleistocene (Schaefer et al., 2016, Nature). Sea-level reconstructions further support the notion that during Marine Isotope Stage 11 and the Pliocene, large parts of Greenland could have been disintegrated (Dutton et al., 2015, Science).

**West Antarctic Ice Sheet:**
Different processes make the West Antarctic Ice Sheet susceptible to tipping dynamics. Since large parts of West Antarctica are marine basins, changes in the ocean are key in driving the evolution of the ice sheet. The Marine Ice Sheet Instability can trigger

self-sustained ice loss where the ice sheet is based below sea-level on retrograde sloping bedrock (Schoof, 2007, J. Geophys. Res.-Earth). This destabilising mechanism is possibly already underway in the Amundsen Sea region (Favier et al., 2014, Nat. Clim. Change; Joughin et al., 2014, Science). Once triggered, a single local perturbation via increased subshelf melting in the Amundsen region could lead to wide-spread retreat of the West Antarctic Ice Sheet (Feldmann & Levermann, 2015, Proc. Natl. Acad. Sci.). Further, a recent study shows strong hysteresis behaviour for the whole Antarctic Ice Sheet, identifying two major thresholds which lead to a destabilisation of West Antarctica around 2°C of global warming, and large parts of East Antarctica between 6–9°C of global warming (Garbe et al., 2020, Nature). It is likely that the West Antarctic Ice Sheet has experienced brief but dramatic retreats during the past five million years (Pollard & DeConto, 2009, Nature). Prior collapses have been suggested from deep-sea-core isotopes and sea-level records (Gasson, 2016, Proc. Natl. Acad. Sci.; Dutton et al., 2015, Science; Pollard & DeConto, 2005, Glob. Planet. Change).

Therefore, we argue that the main dynamics of the ice sheets, even though on time scales of centuries to millennia, can be modelled with Eq. (1).

**AMAZON RAINFOREST:**

Conceptual models of the Amazon identified multi-stability between rainforest, savannah and treeless states, leading to hysteresis (Staal et al., 2016, Ecosystems; Staal et al., 2015, Ecol. Complex.; Van Nes et al., 2014, Glob. Change Biol.). This hysteresis has been found to be shaped by local-scale tipping points of the Amazon rainforest and resilience might be diminished under climate change until the end of the 21st century (Staal et al., 2020, Nat. Commun.). More complex dynamic vegetation models also found alternative stable states of the Amazon ecosystem (Oyama & Nobre, 2003, Geophys. Res. Lett.) and suggest that rainforest dieback might be possible due to drying of the Amazon basin under future climate change scenarios (Nobre et al., 2016, Proc. Natl. Acad. Sci.; Cox et al., 2004, Their. Appl. Climatol.; Cox et al., 2000, Nature). Observational data further supports the potential for multi-stability of the Amazon rainforest (Ciemer et al., 2019, Nat. Geosci.; Hirota et al., 2011, Science; Staver et al., 2011, Science). While it remains an open question whether the Amazon has a single system-wide tipping point, the projected increase in droughts and fires (Malhi et al., 2009, Proc. Natl. Acad. Sci.; Cox et al., 2008, Nature) is likely to impact the forest cover on a local to regional scale, which might spread to other parts of the region via moisture-recycling feedbacks (Zemp et al., 2017, Nat. Commun.; Zemp et al., 2014, Atmospheric Chem. Phys.; Aragão, 2012, Nature). It is important to note that in contrast to the ice sheets and ocean circulation, the rainforest is able to adapt to changing climate conditions to a certain extent (Sakschewski et al., 2016, Nat. Clim. Change). However, this adaptive capacity might still be outpaced if climate change progresses too rapidly (Wunderling et al., 2020, in review). A dieback of the Amazon rainforest has been found under a business-as-usual scenario (Cox et al., 2004, Their. Appl. Climatol.), which would be equivalent to a global warming of more than 3°C

**above pre-industrial levels (3.5–4.5 C (see also Schellnhuber et al., 2016, Nat. Clim. Change)), mainly due to more persistent El-Niño conditions (Betts et al., 2004, Theor. Appl. Climatol.).**

**Therefore, we also argue that the Amazon rainforest can be modelled with an equation of type Eq. (1).**

Issue (ii) is dealt with in lines 167-176 and Table 2, but this by far insufficient and (apart from the ENSO-AMOC connection) without any references to model/observation results. One cannot simply refer only to Kriegler et al., as that assessment was very rough (Fig. 2 in their paper) and more than 10 years old.

**We are very thankful for the reviewers comment since we agree that a better understanding of the interaction processes between the tipping elements does improve our work significantly and also yields a better motivation for the interactions between the tipping elements. Therefore, we supply each interaction pair in our set of four tipping elements with existing literature references. The references motivate why the respective link has a stabilising, destabilising or unclear effect on the influenced tipping element. However, a direct interaction strength between different tipping elements as it would be necessary for our conceptual model cannot be extracted from these literature sources listed. Therefore, we propagate all uncertainties in our large scale Monte Carlo ensemble. Please see below for an explanation of each of the interaction pairs between the four investigated tipping elements or Section 2.2 in ll 194-274 in the manuscript for the changes in the revised manuscript. This means that, although the expert elicitation in Kriegler et al. (2009, Proc. Natl. Acad. Sci.) was rough, the additional literature sources support and refine the results from an early expert elicitation.**

1) **Greenland Ice Sheet (GIS) → AMOC: Increasing freshwater input from enhanced melting of the Greenland Ice Sheet can lead to a weakening of the AMOC, as supported by observations, paleo evidence as well as modelling studies (Caesar et al., 2018, Nature; Robson et al., 2014, Nat. Geosci.; Driesschaert et al., 2007, Geophys. Res. Lett.; Jungclaus et al., 2006, Geophys. Res. Lett.; Rahmstorf et al., 2005, Geophys. Res. Lett.). Between 1992 and 2018, the Greenland Ice Sheet has lost around 3900+-342 Gt of ice (Shepherd et al., 2020, Nature). The ice loss has strongly accelerated in recent years (Sasgen et al., 2020, Communs. Earth & Environ.), and Greenland has been subject to several extreme melt events in the past decade alone (Tedesco & Fettweis, 2020, The Cryosphere; Nghiem et al., 2012, Geophys. Res. Lett.; Tedesco et al., 2011, Environ. Res. Lett.). At the same time, an AMOC weakening of 15% (3+-1 Sv) has been observed since the 1950s (Caesar et al., 2018, Nature). This weakening has at least partially been attributed to freshwater influx into the North Atlantic deep water formation regions due to enhanced melting from Greenland. Paleoclimatic records further suggest that the AMOC could exist in multiple stable states, based on observed temperature**

changes associated with meltwater influx into the North Atlantic (Blunier and Brook, 2001, Science; Dansgaard et al., 1993, Nature). Therefore, it is very likely that a tipping of the Greenland Ice Sheet would lead to a destabilization of the AMOC (see Fig. 1).

2) AMOC → GIS: Reversely, if the AMOC weakens, leading to a decline in its northward surface heat transport, Greenland might experience cooler temperatures (e.g. Jackson et al., 2015, Clim. Dyn.; Timmermann et al., 2007, J. Climate; Stouffer et al., 2006, J. Climate), which would have a stabilizing effect on the ice sheet. With the global climate model HadGEM3, it has been shown that temperatures in Europe could drop by several degrees if the AMOC collapses, regionally up to 8 C (Jackson et al., 2015, Clim. Dyn.). A cooling trend in sea surface temperatures (SST) over the subpolar gyre, as a result of a weakening AMOC, has been confirmed by recent reanalysis and observation data (Caesar et al., 2018, Nature; Jackson et al., 2016, Nat. Geosci.; Frajka-Williams, 2015, Geophys. Res. Lett.; Robson et al., 2014, Nat. Geosci.). This "fingerprint" translates a reduction in overturning strength by 1.7 Sv per century to 0.44 K SST-cooling per century (Caesar et al., 2018, Nature). AMOC regime shifts between weaker and stronger overturning strength during the last glacial period have been associated with large regional temperature changes in Greenland, for example during Dansgaard-Oeschger or Heinrich events (Barker and Knorr, 2016, PAGES). Moreover, there is paleoclimatic evidence from 3.6 million years ago that a weaker North Atlantic current as part of the AMOC fostered Arctic sea-ice growth which might have preceded continental glaciation in the northern hemisphere at that time (Karas et al., 2020, Glob. Planet. Change). Based on these findings we assume that a weakening of the AMOC would have a stabilising effect on the Greenland Ice Sheet (see Fig. 1).

3) West Antarctic Ice Sheet (WAIS) → AMOC: It remains unclear whether increased ice loss from the West Antarctic Ice Sheet has a stabilizing and destabilizing effect on the AMOC (see Fig. 1). Swingedouw et al.(2009) (Swingedouw et al., 2009, Clim. Dyn.) identified different processes based on freshwater hosing experiments into the Southern Ocean, which could be associated with a melting West Antarctic Ice Sheet. Using the EMIC LOVECLIM1.1, the authors revealed effects, some of which would enhance and others would weaken the AMOC strength: (i) First, deep water adjustments are observed. This means that an increase of the North Atlantic Deep Water formation is observed in response to a decrease in Antarctic bottom water production due to the conducted hosing experiment. This mechanism has been termed the so-called bipolar ocean seesaw. (ii) Second, salinity anomalies in the Southern Ocean are distributed to the North Atlantic, which dampens the North Atlantic DeepWater formation (compare to Seidov et al., 2005, Glob. Planet. Change). (iii) Third, the North Atlantic Deep Water formation is enhanced by southern hemispheric wind increase in response to a southern hemispheric

cooling. The reason for this wind increase is the risen meridional temperature gradient between a cooler Antarctic region (due to the hosing experiment) and the equator. This effect has been termed the Drake Passage effect earlier (Toggweiler & Samuels, 1995, Deep Sea Res. Part I Oceanogr. Res. Pap.). Overall, the first and the third mechanism tend to strengthen the AMOC, while the second process would rather lead to a weakening of the AMOC. The specific time scales and relative strengths of these mechanisms is as of yet unclear (Swingedouw et al., 2009, Clim. Dyn.). In a coupled ocean-atmosphere model, a slight weakening of the AMOC was detected for a freshwater input of 1.0 Sv in the Southern Ocean over 100 years (Seidov et al., 2005, Global Planet. Change). However, other studies suggest a stabilisation of the AMOC if influenced by freshwater input from the West Antarctic Ice Sheet due to the effects from the bipolar ocean seesaw by decreasing Antarctic Bottom Water formation as described above (Swingedouw et al., 2008, Geophys. Res. Lett.). Therefore, the direction of this interaction pathway is unclear (see Fig. 1).

4) AMOC → WAIS: The interaction from the AMOC to the West Antarctic Ice Sheet is destabilising (see Fig. 1). In case the AMOC shuts down, sea surface temperature anomalies could appear since the northward heat transport is diminished significantly. This could then lead to a warmer south and colder north, as observed in modelling studies (Weijer et al., 2019, J. Geophys. Res.-Oceans; Timmermann et al., 2007, J. Climate; Stouffer et al., 2006, J. Climate; Vellinga & Wood, 2002, Climatic Change). A model intercomparison study for EMICs and AOGCMs found a sharp decrease of surface air temperatures over the northern hemisphere, while a slight increase over the southern hemisphere and around the Antarctic Ice Sheet has been observed (Stouffer et al., 2006, J. Climate). In their study (Stouffer et al., 2006, J. Climate), a forcing of 1.0 Sv has been applied to the northern part of the North Atlantic Ocean. Therefore, we set this link as destabilising (see Fig. 1).

5) GIS → WAIS & WAIS → GIS: The interaction between the Greenland and the West Antarctic Ice Sheet can be regarded as mutually destabilising, however, with a different magnitude (see Fig. 1). It is a well-known phenomenon from tidal changes that grounding lines of ice sheets are varying (e.g. Sayag & Worster, 2013, Geophys. Res. Lett.). Therefore, the Greenland Ice Sheet and the West Antarctic Ice Sheet could influence each other by sea level rise if one or the other cryosphere element would melt. Gravitational, but also elastic and rotational impacts would then enhance the sea level rise in case one of the huge ice sheets would melt first since then only the other ice sheets exerts strong gravitational forces (Kopp et al., 2010, Clim. Change; Mitrovica et al., 2009, Science). The impact of this effect would be higher if Greenland becomes ice free earlier than the West Antarctic Ice Sheet because many marine terminating ice shelves are located in West Antarctica, but the interaction destabilises in both directions (see Fig. 1).

**6) AMOC → AMAZ (Amazon rainforest): Lastly, the interaction between the AMOC and the Amazon rainforest is set as unclear (see Fig. 1). It is suspected that the intertropical convergence zone (ITCZ) would be shifted southward in case the AMOC collapses. This could cause large changes in seasonal precipitation on a local scale, and could as such have strong impacts on the Amazon rainforest (Jackson et al., 2015, Clim. Dyn.; Parsons, 2014, Geophys. Res. Lett.). In the Earth system model ESM2M, it has been found that a strongly suppressed AMOC, through a 1.0 Sv freshwater forcing, leads to drying over many regions of the Amazon rainforest (Parsons, 2014, Geophys. Res. Lett.). However, some regions receive more rainfall than before. On a seasonal level, the wet season precipitation is diminished strongly, while the dry season precipitation is significantly increased (Jackson et al., 2015, Clim. Dyn.; Parsons, 2014, Geophys. Res. Lett.). This could have consequences for the current vegetation that is adapted to this partially strong seasonal precipitation. But overall, it remains unclear whether the influence from a tipped AMOC to the precipitation in South America has a reducing or increasing influence. Instead, it might differ from locality to locality and is set as unclear in our study (see Fig. 1).**

**In the supplementary material, we provide the same discussion for the interactions that include ENSO (see *Structural sensitivity analysis including ENSO* of the supplementary material).**

Minor:

1. l123-124: A collapse in the AMOC only affects the position of the Hopf bifurcation. So clarify what is meant here.

**This is true. We meant that a collapsing AMOC can trigger a critical transition of ENSO (see e.g. Dekker et al., 2018 Earth Syst. Dynam.). To avoid misunderstandings, we have rephrased this sentence in the manuscript (see ll. 430-431).**

2. l187-188: Current GCMs (in particular with high resolution ocean models) can adequately resolve nonlinear behavior in ENSO.

**We agree with the reviewer and removed this statement from our manuscript. However, in terms of interacting tipping elements, we think state-of-the-art GCMs might not always be the best choice for studies such as this, since large ensembles computed over very long times would be required. Therefore, computational constraints might hinder such a wide analysis as it is done in this work, a point that was also raised for instance in Wood et al. (2019, Clim. Dynam.). Still, if possible, it would be great and highly desirable to investigate some or all of the interactions with GCMs or EMICs in the future (see ll. 305-306).**

3. The x-axis label in Fig. 2 is not readable (at least in my .pdf file). Please adapt.

**Thanks for letting us know. In our version, the x-axes labels are visible [panel a) GMT; panel b) Model time (a.u.)]. We now supply this figure as a high-resolution .png file instead of a .pdf file (see Fig. 2). Please let us know if the problem persists.**

4. In Fig. 3, there is a small transition in ENSO which is also reflected in the WAIS response. However, it stays within the baseline regime for ENSO. What is this small transition and what is causing it?

**Since we now exclude ENSO from our analysis in the main manuscript, we exclude these timelines from our analysis and replace them by timelines without ENSO (see new Fig. 3). However, for completeness, we attach this figure here and explain why there is a small transition in ENSO and WAIS: in panel c), for increases of the global mean temperature of 1.9°C or above for this particular choice of parameters, there is a critical transition of AMOC into the tipped state. Since there is a positive interaction link from AMOC to ENSO (see Fig. S2), there is also a small increase of the state in ENSO, that is, however, not sufficient to tip ENSO over. Furthermore, the state of ENSO is positively feeding back to the state of WAIS (see Fig. S2) such that this pattern is pushed forward to the state of WAIS.**

[Figure]

5. I miss a discussion on the time of transition and the delayed effect one transition has on the occurrence of another one. Here the ice sheets play a dominant initiator role simply because their temperature threshold is lowest. However, it takes a significant amount of time for subsequent meltwater (and sea level) to affect other tipping elements. Of course, it is not in the approach followed in the paper but it is relevant to discuss in the last section.

**We appreciate this comment since such a discussion was indeed missing in the discussion of our manuscript. Therefore, we mention the significant time delay that emerges from the transitions of the cryosphere components that might tip themselves only on the order of centuries up to millennia (see ll 513-520).**

Finally, the software used for the results in the paper should be made publicly available (e.g. through github) so other researchers can check the computations.

**We agree that it would help other researchers when the code is published for the construction of the Monte Carlo ensemble as well as the computation of the tipping events. Thus, we created a github repository that explains the software package (PyCascades, doi: 10.5281/zenodo.4153102) in detail and also contains a folder with the climate tipping elements. At the end of the main manuscript, we supply a "Code**

**Availability Statement", where we refer to this repository that also includes a doi (see ll 525-526).**

---

## Author Response (AR3)

**Dear Michel Crucifix, Dear Reviewer,**

**We are grateful for the appreciation of our revision by the reviewer and are thankful for the additional comments. We have reworked our manuscript according to the suggestions of the reviewer and have addressed all remaining comments.**

**Please find below a detailed point-by-point response. The changes in the manuscript are marked in blue. We are grateful for the opportunity to further improve our manuscript and are looking forward to further feedback.**

**Sincerely yours,**
**Nico Wunderling, Jonathan F. Donges, Jürgen Kurths, Ricarda Winkelmann**

It is a pleasure to see that the authors have taken the time to produce a much better paper than the previous versions. Section 2 now contains a much better justification of the methodology and the coupling between the tipping elements. Also the section 3.4 on including ENSO is now much better positioned. I still have some relatively minor comments, and I hope the authors will consider these to improve the manuscript.

**We appreciate the positive evaluation of our substantially revised manuscript. We have also considered all minor points, please see below.**

1. l28: It is better to use Global Mean Surface Temperature than Global Mean Temperature, because otherwise this can be confused with a volume average, and this is what the authors use.

**Yes, that should be clarified, see ll 27 and ll 272-273 of the manuscript.**

2. l39: Specify better where 'this' refers to.

**We meant that it is unclear how the interactions between the tipping elements would affect the stability of the overall climate system. We have rewritten the according sentence, see ll 38.**

3. l73: the Marine Ice Sheet Instability -> local Marine Ice Sheet instabilities (these are local instabilities related to the topographic bottom slope)

**Thanks for pointing us here. We agree and have rewritten this passage, see ll 69-70.**

4. l77: the influence of Greenland melt water on the AMOC is relatively small; the weakening is mainly due to a changing surface buoyancy forcing. Many CMIP5/6 models show this decline even when there is no melt water input.

**Thanks, this is a good point. We have cited a recent paper, which researches the impact of surface buoyancy on the overturning strength of the AMOC in CMIP5 models (Levang and Schmitt, 2020, J. Climate). We also quoted recent evidence that the AMOC is currently at its weakest state since centuries (Caesar et al., 2021, Nature Geoscience). The changes can be found in ll 75-78.**

5. l93: Following the introduction, in -> In
6. caption Figure 1, l6: initiates -> initiates cascades
7. l120: c_{i 1,2} -> c_{i}^{1,2} or c_{i}^{\pm}

**We thank the reviewer for these corrections and have changed the respective sentences in the manuscript (see ll 92, ll 119 and caption of Fig. 1).**

8. l145: In the Mecking et al. 2016 paper a weak AMOC state is found over a few hundred years. However, the pattern of the AMOC does not correspond to a collapsed state, so this is not really evidence of a multiple equilibrium regime.

9. l140-l151: It is much more convincing to cite papers where explicit bifurcation diagrams have been computed for global ocean models, e.g. Huisman et al., JPO, 40, 551, (2010).

**The reviewer is right and we are thankful for this additional reference. We have checked whether all cited model studies explicitly show a hysteresis in their computations (and removed the Mecking et al. (2016) reference). See manuscript ll 142-144.**

10. l150: AMOC -> AMOC is

**Thanks for catching this typo.**

11. l201-212: The effect of Greenland melt water on the AMOC in the present-day climate is considered to be weak, so I would recommend to restrict to the paleoclimate `evidence' here. If one forces a climate model with realistic melt water fluxes, the response of the AMOC cannot be distinguished from the intrinsic variability of the AMOC.

**We agree and have shortened the respective lines and restricted the evidence of multiple AMOC states to paleoclimatic evidence and modelling studies, see ll 200-205.**

12. l233: conducted hosing experiment -> the release of freshwater in the Southern Ocean.

**We have rephrased the sentence, see ll 226-227.**

13. l244: `stabilization' is confusing here. Probably it is meant that the AMOC amplitude increases.

**It is meant that the AMOC overturning strength would remain at a certain level (or even slightly increase). We have clarified this in the manuscript, see ll 237.**

14. l278, l280, equation (3): it is $\Delta T_{limit,i}$ instead of $T_{limit,i}$, in correspondence with Table 1.

**It should indeed read $T_{limit, i}$ since a specific value of the critical temperature $T_{limit, i}$ is drawn in each member of our Monte Carlo simulation. And the limits between which the critical temperature is uniformly drawn is $\Delta T_{limit, i}$. Those limits are given in Table 1. To avoid confusion, we have explicitly mentioned that in ll 274-275 of the manuscript.**

15. l321-322: model years -> time (`model years' always have units)

**We agree and have replaced model years by time, see ll 316-317.**

16. l355: `randomly' is not specific enough. What distribution is used?

**We agree that this should be noted in the manuscript (see ll 351). We base the Monte Carlo ensemble on a continuous uniform distribution between the respective limits of the drawn parameter values.**

17. l381: 'likely due to' is too vague. For such a simple model, this can be precisely determined so please do so.

**This is correct and we checked this: we can omit the word 'likely' from this sentence.**

18. l431: the multiple equilibria view on ENSO has long been abandoned (it is also not in the Dekker et al. (2018) paper), so please omit it here.

**Indeed, it is better to say that a Hopf bifurcation in ENSO has been observed in modelling studies from the literature (e.g. Dekker et al. (2018), ESD) in case the ENSO-component is forced by a tipping AMOC-component. We rephrased the according sentence (see ll 428).**

19. l499: start a new paragraph

**We have started a new paragraph at this line (see ll 498).**

**List of additional references in the manuscript:**

1. **Caesar, L., McCarthy, G.D., Thornalley, D.J.R., Cahill, N. and Rahmstorf, S.: Current Atlantic Meridional Overturning Circulation weakest in the last millennium. Nat. Geosci., 1-3, 2021.**
2. **Huisman, S.E., Den Toom, M., Dijkstra, H.A. and Drijfhout, S.: An indicator of the multiple equilibria regime of the Atlantic Meridional Overturning Circulation, J. Phys. Oceanogr., 40, 551-567, 2010.**
3. **Levang, S.J. and Schmitt, R.W.: What Causes the AMOC to Weaken in CMIP5?. J. Climate, 33, 1535-1545, 2020.**